# Beyond RLHF: A Theoretical Framework of Alignment as Distribution Learning

## Abstract

Alignment via reinforcement learning from human feedback (RLHF) has become the dominant paradigm for controlling the quality of outputs from large language models (LLMs). However, the standard RLHF objective lacks formal justification and incentivizes degenerate, deterministic LMs in the asymptotic regime. We ask under what assumptions can we derive RLHF or other novel objectives with rigorous learning-theoretic guarantees, without relying on an *a priori* notion of reward maximization. To this end, we reframe alignment as *distribution learning* from pairwise preferences, formalizing our approach with a probabilistic assumption describing how preferences reveal information about the target (oracle) LM. This leads us to propose three principled alignment objectives: preference maximum likelihood estimation, preference distillation, and reverse KL minimization. We prove that all three approaches enjoy strong non-asymptotic $O(1/n)$ convergence to the target LM, naturally avoiding degeneracy. In particular, reverse KL highly resembles the RLHF objective, providing strong justification for RLHF with a minor correction. Furthermore, our theory confirms the empirical finding that on-policy objectives (e.g., RLHF) often outperform likelihood-style objectives (e.g., DPO). Finally, we empirically show that our proposed methods consistently matches or outperforms baselines across various tasks and models.

## 1 Introduction

Alignment refers to the task of improving the quality of responses (e.g., helpfulness and harmlessness) generated from large language models (LLMs) via human preferences (Bai et al., 2022; Ouyang et al., 2022) and has become the de facto final step in LLM training. The first method introduced for alignment is Reinforcement Learning from Human Feedback (RLHF) (Christiano et al., 2017; Stiennon et al., 2020), which trains a reward model $R$ from pairwise preferences and then optimizes a policy $\pi$ (i.e., language model) that maximizes the reward via reinforcement learning (RL):

$$\max_{\pi} \; \mathbb{E}_{x\sim\mathcal{D},a\sim\pi(x)} \left[ \hat{R}(x,a) \right] - \beta \, \mathbb{E}_{x\sim\mathcal{D}} \left[ \mathrm{KL}(\pi(x)\|\pi_0(x)) \right] \tag{1}$$

where $\hat{R}$ is a reward function learned from preference data, $\mathcal{D}$ is the prompt distribution, $\pi(x)$ is the policy $\pi$'s response distribution for prompt $x$, $\mathrm{KL}(p\|q)$ is the Kullback-Leibler divergence, $\pi_0$ is a supervised fine-tuned reference LLM, and $\beta > 0$ is the regularization strength.

The RLHF objective is central to various practical algorithms and has fundamentally shaped how researchers think about alignment. For example, DPO (Direct Policy Optimization) can be viewed as reformulating RLHF so the objective consists of simple likelihood ratio terms rather than relying on on-policy responses (Rafailov et al., 2023), while $\Psi$PO extends RLHF by generalizing $R(x,a)$ to a $\Psi$-transformation of the preference probability (Azar et al., 2024). Even theoretical analyses of alignment algorithms often treat the RLHF objective or its variants as the ultimate learning-theoretic goal, aiming to establish statistical convergence guarantees (e.g., regret bounds) for its solution (Zhan et al., 2024; Xiong et al., 2024; Zhang et al., 2024; Huang et al., 2025; Xie et al., 2025b). Certainly, the RLHF objective has proven useful, and one may argue that it is a sensible objective.

However, the RLHF objective fundamentally lacks justification from a learning-theoretic viewpoint. While maximizing rewards *informally* sounds reasonable, it is not clear what it *formally* means because the problem definition of alignment does not involve rewards at all! This *a priori* dependency on reward manifests in the following issue: (1) is a standard ML objective of the form 'loss +

Table 1: Summary of our proposed methods and theoretical guarantees. In each section, we draw parallels to existing approaches such as DPO and REBEL (Gao et al., 2024).

| Distribution Learning | Related to | Reward Model | Requires RL Training | Objective | Forward KL Guarantee |
|---|---|---|---|---|---|
| **Preference MLE** (Sec. 3) | DPO | Not Used | No | Eq. 4 | $O(1/n)$ (Thm. 4) |
| **Preference distillation** (Sec. 4) | REBEL | Required | No | Eq. 11 | $O(1/n)$ (Thm. 6) |
| **Reverse KL** (Sec. 5) | RLHF | Required | Yes | Eq. 16 | $O(1/n)$ (Thm. 7) |

regularizer' where the loss (on-policy reward) learns from data and the regularizer exists to leverage an available starting model. With sufficient data, the regularization strength $\beta$ should reduce, and in the asymptotic regime, should become zero to ensure sufficient learning. In this sense, asymptotically, the RLHF objective faces a dilemma: it must either tend to zero regularization so that the solution to (1) becomes deterministic, or use nonzero regularization and hinder the learning process.

Then, how should one formulate effective learning objectives? A reliable strategy is to posit a probabilistic data-generating process and apply principles such as maximum likelihood estimation. This paradigm is widely used in classification (e.g., logistic regression), topic modeling (e.g., latent Dirichlet allocation), and generative models (e.g., variational autoencoder). This offers two key benefits: (i) explicit modeling assumptions which often help understand model behavior, and (ii) rigorous learning-theoretic guarantees such as consistency – convergence to the target model as the sample size grows. Perhaps surprisingly, a fully probabilistic framework for alignment remains largely unexplored; existing works *a priori* assume some reward function is to be maximized, and only study convergence to its optimum. See Appendix A for a detailed discussion.

**Our contributions.** In this paper, we move beyond blindly taking the RLHF objective as the ultimate goal. We propose a novel **distribution learning** perspective based on a purely probabilistic approach, which does not rely on an *a priori* notion of reward maximization. Instead, we posit that there exists a target (oracle) language model (LM) $\pi^*$ and explicitly model how information about $\pi^*$ is revealed through preference feedback. Intuitively, $\pi^*$ must *assign a higher probability to the preferred response*, which we encode as the assumption

$$\mathbb{P}(a \succ b \mid x) = \frac{\pi^*(a \mid x)^\gamma}{\pi^*(a \mid x)^\gamma + \pi^*(b \mid x)^\gamma} \qquad (2)$$

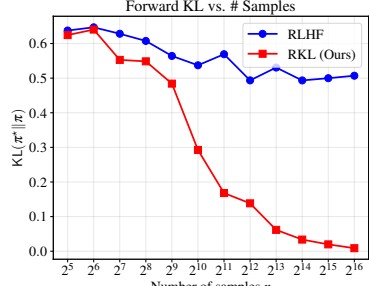

Figure 1: RLHF vs. RKL.

for some $\gamma > 0$ where $a \succ b$ means the response $a$ is preferred over $b$. This is a Bradley-Terry (BT) model (Bradley & Terry, 1952) with preference score proportional to tilted response likelihood from $\pi^*$. The main difference from existing works that use the BT model is that our assumption says that the preference model is *explicitly* a language model. This is in stark contrast to DPO which starts from the RLHF formulation and leverages the all-policy assumption to realize that there is a *secret* relationship between the reward model (or preference model) and the LM (Rafailov et al., 2023). Figure 1 shows that under our preference model, RLHF with tuned $\beta$ for each sample size $n$ indeed suffers from inconsistency (i.e., does not converge to the true $\pi^*$) where as our proposed correction (Reverse KL, see Section 5) is consistent.

Our simple assumption leads to various training objectives whose solutions provably converge to $\pi^*$ with respect to metrics such as KL divergence. Specifically, we propose the following three algorithms (summarized in Table 1):

- **PMLE** (Preference Maximum Likelihood Estimate; Section 3): This objective maximizes the likelihood of the preference model (2), subject to reverse KL regularization w.r.t. a reference policy $\pi_0$. Similarly to DPO, it is relatively straightforward to optimize.
- **Preference distillation** (Section 4): By directly estimating the expected preference from a learned reward model, the MLE can be rewritten as distilling the preference distribution into a language model. Unlike existing reward distillation (Fisch et al., 2025; Gao et al., 2024), this formulation is explicitly derived from the Bradley-Terry model (2).
- **Reverse KL** (RKL; Section 5): Since our goal is distribution learning, it is also natural to optimize the reverse KL divergence $\mathbb{E}_x[\text{KL}(\hat{\pi}(x)\|\pi^*(x))]$. Although $\pi^*$ is unknown, its unnormalized form can be estimated from (2) with a shallow network, which amounts to learning a reward model

in RLHF. Plugging in our estimate of $\pi^*$ in the reverse KL along with a KL regularizer yields a variant of the RLHF objective that has an additional entropy term, effectively smoothing the prior $\pi_0$. This framework thus offers a new statistical grounding for RLHF.

- **Theoretical guarantees**: For all three algorithms, we prove upper bounds on the forward KL error of the form: $\mathbb{E}_x[\mathrm{KL}(\pi^*(x)\|\hat{\pi}(x))] \leq O(1/n)$ where $n$ is the size of the preference dataset.

Our guarantees are non-asymptotic and first-of-its-kind for learning a distribution from pairwise feedback, to the best of our knowledge (cf. Dumoulin et al., 2023). A full discussion of related works is deferred to Appendix A. All proofs are deferred to Appendix C. We complement our theory with experiments showing that our methods consistently outperform baseline win-rates in TL;DR summarization and generate more preferred responses in general chat experiments.

## 2 PRELIMINARIES

**Alignment as distribution learning.** Let $\mathcal{X}$ and $\mathcal{A}$ be the space of prompts and responses, respectively, and let $\mathcal{D} \in \Delta(\mathcal{X})$ be a fixed distribution over prompts. We define a language model (LM) as a function or policy $\pi : \mathcal{X} \to \Delta(\mathcal{A})$ determining a collection of conditional (i.e., contextual) distributions $\pi(\cdot \mid x)$, which we also denote more simply as $\pi(x)$.[1] We view alignment as learning these distributions from pairwise preference feedback, drawn from a model explicitly depending on $\pi^*$, the ideal (target) LM we wish to learn. Hence given a class $\Pi$ of LMs, our ultimate goal is to find $\hat{\pi} \in \Pi$ that is as close as possible to $\pi^*$ w.r.t. a suitable measure of distance between distributions.

**Our preference model.** Let $\mu$ be the LM used for generating responses to be preference-labeled; this could be a reference LLM or simply an existing dataset. We are given a preference dataset $D_n = \{(x, a^+, a^-)\}$ of $n$ independent samples where $x \sim \mathcal{D}$ is a prompt, $a^+$ is a preferred response, and $a^-$ is a dispreferred response. We assume that, given $x$, the pair $(a^+, a^-)$ is sampled by drawing $a, b \sim \mu(x)$ independently and then sampling a preference from $\mathbb{P}_*(a \succ b \mid x)$, where

$$\mathbb{P}_* := \mathbb{P}_{\pi^*} \quad \text{and} \quad \mathbb{P}_\pi(a \succ b \mid x) := \frac{\pi(a \mid x)^\gamma}{\pi(a \mid x)^\gamma + \pi(b \mid x)^\gamma}, \tag{3}$$

followed by setting $(a^+, a^-) = (a, b)$ if $a$ is preferred over $b$ and $(a^+, a^-) = (b, a)$ otherwise. The value of $\gamma$ determines the extent to which differences in the response probabilities under policy $\pi$ are accentuated or attenuated. In practice, $\gamma$ is a hyperparameter typically set as $0 < \gamma < 1$.

**Is our preference model too strong?** One may wonder if (2) is too restrictive; we claim that it is not. Our assumption (i) takes the BT model $\mathbb{P}(a \succ b|x) = p^*(a|x)/(p^*(a|x) + p^*(b|x))$ for some underlying preference $p^*$ and then (ii) connects $p^*$ with the target LM $\pi^*$. For (i), while the BT model has drawbacks such as not allowing cyclic preferences, it is conventional in many prior works (Christiano et al., 2017; Stiennon et al., 2020). Also, our distribution learning perspective can be applied to derive objectives based on more general preference models. More importantly, (ii) is the crucial link allowing us to derive guarantees for learning $\pi^*$. Previous theoretical studies provide convergence guarantees for RLHF or other reward-maximizing objectives (Zhan et al., 2024; Xiong et al., 2024; Zhang et al., 2024; Huang et al., 2025; Xie et al., 2025b) without justifying why a certain reward construction must be optimized in the first place. In contrast, the main point of our work is to identify under which assumptions such objectives can be justified. Indeed, we will see that objectives derived from our framework often resemble baselines (DPO, RLHF, REBEL), so that we are essentially making the hidden assumptions of existing methods explicit!

**Theoretical setup.** We call $R_\pi(x, a) := \gamma \ln \pi(a \mid x)$ the *reward* induced by $\pi \in \Pi$.[2] The centered reward is defined as $\bar{R}_\pi(x, a) := R_\pi(x, a) - \mathbb{E}_{a \sim \mu(x)}[R_\pi(x, a) \mid x]$. As with $\mathbb{P}_*$, we write $R_* := R_{\pi^*}$ and $\bar{R}_* := \bar{R}_{\pi^*}$. Finally, we denote $\Delta \bar{R}_\pi := \bar{R}_\pi - \bar{R}_*$.

Our main assumptions, standard in the alignment literature (Zhan et al., 2024; Xie et al., 2025a; Zhang et al., 2025; Agarwal et al., 2025; Huang et al., 2025), are as follows:

**Assumption 1** (Realizability). $\pi^* \in \Pi$ for a finite policy class $\Pi$.

**Assumption 2** (Boundedness). There exists $R > 0$ such that $|\bar{R}_\pi(x, a)| \leq \gamma R$ for all $\pi \in \Pi$.

---

[1] This definition can also cover unconditional distributions by introducing a member in $\mathcal{X}$ as a null prompt.
[2] There is no actual reward in the PMLE scheme; we just call this reward for convenience.

Since the responses $(a^+, a^-)$ are sampled from $\mu(x)$ rather than $\pi^*(x)$, the alignment problem is an instance of offline learning where there is a distribution shift between the observed data versus the target distribution that we aim to have guarantees on. It is thus necessary to introduce a coverage assumption between $\mu$ and the policy class $\Pi$, which is well-studied in the offline RL literature (Agarwal et al., 2019). In particular, we use the following generalized coverage coefficient (Xie et al., 2021; Agarwal et al., 2025).

**Definition 3** (Generalized coverage coefficient). For a policy class $\Pi'$, we denote by $C_{\Pi'} > 0$ the smallest constant satisfying for every $\pi \in \Pi'$,

$$\mathbb{E}_{x \sim \mathcal{D}, a \sim \pi^*(x)} \left[ \Delta \bar{R}_\pi(x, a)^2 \right] \leq C_{\Pi'} \mathbb{E}_{x \sim \mathcal{D}, a \sim \mu(x)} \left[ \Delta \bar{R}_\pi(x, a)^2 \right].$$

This improves upon the all-policy $\ell_\infty$-concentrability $\sup_{\pi \in \Pi} \max_{x,a} \frac{\pi(a|x)}{\mu(a|x)} \leq C'$ (Munos, 2003) as the former can be bounded even if the latter is infinite, depending on $\mathcal{D}$ and the reward class $\Delta \bar{R}$.[3]

# 3 PREFERENCE MAXIMUM LIKELIHOOD ESTIMATION APPROACH

We begin by introducing a maximum likelihood-based objective that can be directly derived from treating alignment as distribution learning from pairwise feedback. Given a preference dataset $D_n = \{(x, a^+, a^-)\}$ as described in Section 2, we wish to estimate $\pi^*$ by finding a policy $\hat{\pi}$ that maximizes the likelihood of observed pairwise preferences under the BT preference assumption (3). Concretely, the negative log-likelihood for each pair $(x, a^+, a^-)$ under a candidate policy $\pi$ is:

$$-\ln \mathbb{P}_\pi(a^+ \succ a^- \mid x) = -\ln \left[ \sigma \left( \gamma \ln \frac{\pi(a^+ \mid x)}{\pi(a^- \mid x)} \right) \right],$$

where $\sigma(z) = 1/(1 + \exp(-z))$ is the logistic sigmoid. Summing over all preference pairs yields

$$\mathcal{L}_{\text{PMLE}}(\pi) = \frac{1}{n} \sum_{(x, a^+, a^-) \in D_n} -\ln \left[ \sigma \left( \gamma \ln \frac{\pi(a^+ \mid x)}{\pi(a^- \mid x)} \right) \right]. \tag{4}$$

By minimizing $\mathcal{L}_{\text{PMLE}}$, we encourage $\pi$ to place higher probability on response $a^+$ relative to $a^-$. Note that in practice, we rarely learn a policy $\pi$ from scratch; instead, we typically optimize a fine-tuned model, referred to as the *reference policy* $\pi_0$. Thus, it is natural to introduce a KL penalty that keeps $\pi$ close to $\pi_0$ for alignment: $\beta \cdot \text{KL}(\pi(x) \| \pi_0(x))$. Putting everything together, our **PMLE** (preference maximum likelihood estimation) objective for distribution learning is

$$\mathcal{L}_{\text{PMLE}, \beta}(\pi) = \frac{1}{n} \sum_{(x, a^+, a^-) \in D_n} -\ln \left[ \sigma \left( \gamma \ln \frac{\pi(a^+ \mid x)}{\pi(a^- \mid x)} \right) \right] + \beta \, \text{KL}(\pi(x) \| \pi_0(x)). \tag{5}$$

**Remark.** Recall that DPO (Rafailov et al., 2023) minimizes the objective

$$\mathcal{L}_{\text{DPO}}(\pi) = \frac{1}{n} \sum_{(x, a^+, a^-) \in D_n} -\ln \left[ \sigma \left( \gamma \ln \frac{\pi(a^+ \mid x)}{\pi(a^- \mid x)} - \gamma \ln \frac{\pi_0(a^+ \mid x)}{\pi_0(a^- \mid x)} \right) \right]. \tag{6}$$

Compared to (5), the DPO objective does not have an explicit regularizer, which could lead to undesirable behaviors if the policy class $\Pi$ is sufficiently expressive. Specifically, Fisch et al. (2025) prove that DPO may converge to a degenerate distribution. Also, Song et al. (2024) show that DPO relies on a strong coverage assumption: if $\pi_0$ does not fully cover the relevant distribution, DPO can produce out-of-distribution responses, making its reward estimates inaccurate. Unlike RLHF, which has a KL term to stay within the support of $\pi_0$, DPO can assign non-zero probability to responses that $\pi_0$ would never select, undermining performance guarantees. In contrast, our PMLE objective (5) incorporates an explicit KL term that effectively circumvents the aforementioned pitfalls.

**Convergence guarantee.** Under the assumptions stated in Section 2, we show the following bound on the forward KL. Throughout the paper, constants depending only on $R$ are hidden.

---

[3]Leveraging pessimism (Gabbianelli et al., 2024; Huang et al., 2025; Zhan et al., 2022) may further improve the coverage coefficient to a single concentrability coefficient that relies on $\pi^*$ rather than the policy class $\Pi$.

**Theorem 4.** *The PMLE estimate $\hat{\pi} = \arg\min_{\pi \in \Pi} \mathcal{L}_{\mathsf{PMLE}}(\pi)$ satisfies with probability at least $1 - \delta$,*

$$\mathbb{E}_{x \sim \mathcal{D}} \left[ \mathrm{KL}(\pi^*(x) \| \hat{\pi}(x)) \right] \lesssim \frac{C_\Pi}{\gamma^2} \cdot \frac{\ln(|\Pi|/\delta)}{n}. \tag{7}$$

The proof, provided in Appendix C.2, is inspired in part by Agarwal et al. (2025, proof of Theorem 3.6), but we leverage Schulman's trick (Schulman, 2020) followed by a quadratic approximation to obtain a $1/n$ rate rather than $1/\sqrt{n}$ that would be obtained when directly following their proof. Also note that the left-hand side of (7) is equivalent to the KL divergence between the induced *joint* distributions on $\mathcal{X} \times \mathcal{A}$: $\mathrm{KL}(\mathcal{D}(x)\pi^*(a \mid x) \| \mathcal{D}(x)\hat{\pi}(a \mid x))$.

We assume $\beta = 0$ here and for all analysis in the sequel for simplicity and to demonstrate that the objective derived from purely considering preference feedback via (3) already suffices to learn the true distribution $\pi^*$. Nonetheless, we posit that starting from a well-aligned $\pi_0$ can result in improved guarantees by mitigating the dependency of constants on $R$, which we leave to future work.

Next, we turn our focus to alignment methods that require an explicit reward model. As per our philosophy, we emphasize that the methods are derived from a distribution learning perspective rather than reward maximization.

## 4 PREFERENCE DISTILLATION APPROACH

Since the popularization of RLHF, the use of reward modeling has become popular in the research community and resulted in various extensions (Christiano et al., 2017). While the main role of the reward model in the RLHF objective (1) is to view alignment as an RL problem, recent studies have attempted to use the reward model for supervised learning losses, i.e., objectives that do not require RL to solve (Guo et al., 2024; Fisch et al., 2025). These efforts can be seen as *distilling information* from the reward model as pointed out by Fisch et al. (2025). The main benefit of these methods is that they can avoid RL algorithms, which are typically slow to converge. While reward model training is an extra burden to perform compared to purely likelihood-based methods such as DPO or our PMLE, the compute cost for doing so is typically quite low because it usually suffices to train a shallow network on top of an existing LLM's frozen torso.

**Reward model.** Due to our preference model (3), learning a reward model $R : \mathcal{X} \times \mathcal{A} \to \mathbb{R}$ is equivalent to learning a language model $\pi$ and then setting $R(x, a) = \gamma \ln \pi(a \mid x)$ up to an additive constant. Conversely, given a reward model $R(x, a)$, we can estimate an LM by

$$\pi(a \mid x) \propto \exp(\gamma^{-1} R(x, a)), \quad \forall x \in \mathcal{X}. \tag{8}$$

Note that this is a model from which sampling is computationally intractable in general. Formally, we assume that we are given a reward model class $\mathcal{R}$ of rewards $R : \mathcal{X} \times \mathcal{A} \to \mathbb{R}$ and learn:

$$\hat{R} = \arg\min_{R \in \mathcal{R}} \frac{1}{n} \sum_{(x, a^+, a^-) \in D_n} -\ln \sigma(R(x, a^+) - R(x, a^-)). \tag{9}$$

This is equivalent to the PMLE objective under (8) but with the constraint $R \in \mathcal{R}$.

**Preference distillation.** One popular method for distilling rewards is the REBEL algorithm (Gao et al., 2024). Motivated by the characterization of the RLHF solution under the all-policy assumption (Rafailov et al., 2023), REBEL aims to extract information from relative reward values of paired responses, enforcing $\ln \frac{\pi(a^+|x)/\pi_0(a^+|x)}{\pi(a^-|x)/\pi_0(a^-|x)} \approx \eta(\hat{R}(x, a^+) - \hat{R}(x, a^-))$ by optimizing a squared loss

$$\frac{1}{n} \sum_{(x, a^+, a^-) \in D_n} \left( \ln \frac{\pi(a^+ \mid x)/\pi_0(a^+ \mid x)}{\pi(a^- \mid x)/\pi_0(a^- \mid x)} - \eta(\hat{R}(x, a^+) - \hat{R}(x, a^-)) \right)^2 \tag{10}$$

where $\eta > 0$ controls the strength of the reward signals. In our assumption, the reward model can be seen as a shifted version of $\gamma \ln \pi^*(a \mid x)$, so we could optimize (10) without the $\pi_0$ terms, replacing $\eta$ by $\gamma^{-1}$. However, the use of squared loss in (10) is not well justified from a statistical perspective, and it is unclear if squared loss should be preferred over any other loss, e.g., absolute loss.

What is then the appropriate error measure? Our framework tells us that learning a reward model amounts to learning a preference model. In other words, we have trained a *preference simulator*: a

non-generative language model estimate $\widetilde{\pi}(a \mid x) \propto \exp(\gamma^{-1}\hat{R}(x,a))$ from which preference can be sampled for any pair of responses as $y \sim \text{Bernoulli}(\mathbb{P}_{\widetilde{\pi}}(a \succ b \mid x))$. Plugging this into PMLE would yield a natural distribution learning objective. However, this process introduces additional randomness which can hinder optimization. Instead, observe that we can evaluate the expectation of the PMLE objective and replace the discrete label $y$ with the *expected* preference

$$\mathbb{P}_{\widetilde{\pi}}(a^+ \succ a^- \mid x) = \frac{\widetilde{\pi}(a^+ \mid x)^\gamma}{\widetilde{\pi}(a^+ \mid x)^\gamma + \widetilde{\pi}(a^- \mid x)^\gamma} = \sigma(\hat{R}(x,a^+) - \hat{R}(x,a^-)).$$

Then, minimizing the log-loss with respect to this synthetic preference leads to:

$$\mathcal{L}_{\text{Distill}}(\pi) := \frac{1}{n} \sum_{(x,a^+,a^-)\in D_n} - \mathbb{P}_{\widetilde{\pi}}(a^+ \succ a^- \mid x) \ln \mathbb{P}_\pi(a^+ \succ a^- \mid x)$$
$$- \mathbb{P}_{\widetilde{\pi}}(a^- \succ a^+ \mid x) \ln \mathbb{P}_\pi(a^- \succ a^+ \mid x). \quad (11)$$

This amounts to minimizing the KL between the binary preference distributions, $\text{Bern}(\mathbb{P}_{\widetilde{\pi}}(a^+ \succ a^-|x))$ and $\text{Bern}(\mathbb{P}_\pi(a^+ \succ a^-|x))$. As in the PMLE (Section 3), in practice we add a KL regularizer:

$$\mathcal{L}_{\text{Distill},\beta}(\pi) := \mathcal{L}_{\text{Distill}}(\pi) + \beta\,\mathbb{E}_{x\sim\mathcal{D}}\left[\text{KL}(\pi(x)\|\pi_0(x))\right]. \quad (12)$$

We remark that the data for reward model training (9) and preference distillation (11) can come from different datasets; our theoretical analysis is easily adapted.

**Convergence guarantee.** The family of (non-generative) language models induced by the reward model class $\mathcal{R}$ is defined as

$$\mathcal{P}_\gamma(\mathcal{R}) := \left\{\pi : \pi(a \mid x) \propto \exp(\gamma^{-1}R(x,a)), \forall a, x \in \mathcal{X} \text{ for some } R \in \mathcal{R}\right\}.$$

**Assumption 5.** The reward-induced LM class $\mathcal{P}_\gamma(\mathcal{R}) \subseteq \Pi$.

This assumption is related to the generator-verifier gap, which informally states that verifying whether a given answer is correct or not is easier than generating a correct answer (Li et al., 2024; West et al., 2024). Such a gap implies that $\mathcal{R}$ is easier to learn than $\Pi$ from a learning-theoretic perspective ($|\mathcal{R}| \ll |\Pi|$), and is speculated to hold for LLMs in practice (Swamy et al., 2025). Assumption 5 can also be justified by the fact that the RM is often built on top of the supervised fined-tuned model's (frozen) torso. Denoting by $C_\mathcal{R} := C_{\mathcal{P}_\gamma(\mathcal{R})}$ the generalized coverage coefficient of the induced subclass, under Assumption 5 it holds that $C_\mathcal{R} \leq C_\Pi$.

**Theorem 6.** *The pref. distill. estimate* $\hat{\pi} = \arg\min_{\pi\in\Pi} \mathcal{L}_{\text{Distill}}(\pi)$ *satisfies with probability at least* $1-\delta$,

$$\mathbb{E}_{x\sim\mathcal{D}}\left[\text{KL}(\pi^*(x)\|\hat{\pi}(x))\right] \lesssim \frac{C_\Pi}{\gamma^2} \cdot \frac{\ln(|\Pi|/\delta)}{n}. \quad (13)$$

See Appendix C.3 for the proof.

**Benefits of distillation.** The above rate $n \gtrsim C_\Pi \ln|\Pi|$ is equal to that of PMLE (7) since we assume we learn $\hat{\pi} \in \Pi$ using responses $D_n$ generated from $\mu$, same as the reward model. However, if we have access to a 'stronger' base model $\pi_0$, it is natural to learn $\hat{\pi}$ using responses from $\pi_0$ instead. In this scenario, we instead obtain a rate of $\gamma^{-2}(C_0 \ln|\Pi| + C_\mathcal{R} \ln|\mathcal{R}|)$, where $C_0$ is a slightly modified coverage coefficient for $\pi_0$ instead of $\mu$. This improves over the PMLE rate if $\pi_0$ is comparatively well-aligned so that $C_0 < C_\Pi$, thus rigorously establishing the benefits of distillation with a strong base model. See Theorem 14 in Appendix C.3 for a formal statement and proof.

## 5 REVERSE KL MINIMIZATION APPROACH

Our two proposed methods both maximize a preference likelihood and ultimately enjoy a guarantee on the forward KL divergence $\mathbb{E}_x[\text{KL}(\pi^*(x)\|\hat{\pi}(x))]$. However, it is also plausible to aim to minimize the reverse KL divergence $\mathbb{E}_x[\text{KL}(\hat{\pi}(x)\|\pi^*(x))]$ to learn the distribution $\pi^*$. Reverse KL has the well-known 'mode seeking' behavior as opposed to 'mode covering' behavior of the forward KL. This mode-seeking behavior tends to find distributions that generate realistic content in image generation and has been preferred in image generative models (Goodfellow et al., 2014; Mao et al., 2019).

In this section, we explore the reverse KL formulation for alignment under our modeling assumption (3), which turns out to be a generalization of the original RLHF framework (1) (Stiennon et al.,

2020; Ouyang et al., 2022). Directly minimizing the reverse KL w.r.t. the target LM $\pi^*$ would yield:

$$\hat{\pi} = \arg\min_{\pi \in \Pi} \mathbb{E}_{x \sim \mathcal{D}} \big[ \mathrm{KL}(\pi(x) \| \pi^*(x)) \big]$$

$$= \arg\min_{\pi \in \Pi} \mathbb{E}_{x \sim \mathcal{D}} \Big[ \mathbb{E}_{a \sim \pi(x)} \big[ -\ln \pi^*(a \mid x) \big] + H(\pi(x)) \Big], \tag{14}$$

where $H(\pi(x))$ is the Shannon entropy of $\pi(x)$. However, this requires rewards of the form $-\ln \pi^*$, which is the very object we are trying to estimate. To solve this issue, we propose to find a plugin estimator from a surrogate class of language models that are easier to train but harder to sample from. Specifically, we determine $\widetilde{\pi} = \arg\min_{\pi \in \mathcal{P}_\gamma(\mathcal{R})} \mathcal{L}_{\mathsf{PMLE}}(\pi)$ (with a suitable regularization), which is equivalent to obtaining $\hat{R}$ via (9) followed by setting $\widetilde{\pi}(a \mid x) \propto \exp(\gamma^{-1} \hat{R}(x, a))$ as before. Then we can plug in our learned $\widetilde{\pi}$ to $\pi^*$ in (14) to arrive at the objective

$$\hat{\pi} = \arg\min_{\pi \in \Pi} \mathbb{E}_{x \sim \mathcal{D}} \Big[ \mathbb{E}_{a \sim \pi(x)} \big[ -\gamma^{-1} \hat{R}(x, a) \big] - H(\pi(x)) \Big]. \tag{15}$$

The normalization constant, which is prohibitive to compute in practice, naturally disappears as we only require relative rewards for optimization. Lastly, we again add a KL regularizer w.r.t. $\pi_0$:

$$\mathcal{L}_{\mathsf{RKL},\beta}(\pi) := \frac{1}{n} \sum_{(x,\cdot,\cdot) \in D_n} -\mathbb{E}_{a \sim \pi(x)} \big[ \hat{R}(x, a) \big] - \gamma H(\pi(x)) + \beta \, \mathrm{KL}(\pi(x) \| \pi_0(x)) \tag{16}$$

where $\beta$ and $\gamma$ control the relative weights of the policy entropy and KL regularizer, respectively. In practice, as in standard RLHF pipelines (Ouyang et al., 2022; Bai et al., 2022), one first fits a reward model $\hat{R}$ to approximate the underlying true reward $R^*$ from pairwise preferences, then applies an RL algorithm (e.g., PPO (Schulman et al., 2017)) to minimize the objective (16).

**Relation to RLHF.** At face value, RKL can be seen as a generalization of RLHF since the RKL objective with $\gamma = 0$ is equal to the RLHF objective. Conversely, under our preference assumption (3), RLHF itself can be interpreted as minimizing a reverse KL in the limit $\gamma \to 0$. Thus, our result on RKL can be viewed as providing *theoretical justification* for the RLHF objective (1) which has been widely viewed as the gold standard for alignment (Stiennon et al., 2020; Bai et al., 2022; Rafailov et al., 2023), *while also providing a minor correction.* Note that such a connection to RLHF may not be surprising given that the max entropy RL can be seen as reverse KL minimization (Ziebart, 2010).

**The dilemma of RLHF.** As discussed in the introduction, RLHF suffers from the following asymptotic dichotomy, assuming $\mu$ has sufficient coverage:

1. **Underfitting** (fix $\beta > 0$ for all $n$): The learned LM is a proper distribution but cannot be too far from the reference policy $\pi_0$.
2. **Degenerate** (take $\beta = \beta_n \downarrow 0$ as $n \to \infty$): The learned LM collapses to a degenerate solution.

Neither is desirable. This phenomenon can also be seen in our toy experiment shown in Figure 1.[4] As demonstrated in Theorem 7, the forward KL with RKL indeed tends to zero as the sample size $n$ increases, whereas that with RLHF decreases slightly but eventually saturates at a non-zero value.

**Convergence guarantee.** With the objective $\mathcal{L}_{\mathsf{RKL}} := \mathcal{L}_{\mathsf{RKL},0}$, we are indeed able to obtain the following guarantee proved in Appendix C.4:

**Theorem 7.** *The reverse KL estimate* $\hat{\pi} = \arg\min_{\pi \in \Pi} \mathcal{L}_{\mathsf{RKL}}(\pi)$ *satisfies with probability at least* $1 - \delta$,

$$\mathbb{E}_{x \sim \mathcal{D}} \big[ \mathrm{KL}(\pi^*(x) \| \hat{\pi}(x)) \big] \lesssim \frac{\ln(|\Pi|/\delta)}{n} + \frac{C_\mathcal{R}}{\gamma^2} \cdot \frac{\ln(|\mathcal{R}|/\delta)}{n}. \tag{17}$$

**Why does reverse KL attain a better bound?** The reverse KL formulation results in an improved upper bound for the *forward* KL that depends on the coverage coefficient of $\mathcal{P}_\gamma(\mathcal{R})$ rather than $\Pi$ as in previous bounds (7), (13). In particular, under Assumption 5, $\ln |\Pi|$ and $C_\mathcal{R} \ln |\mathcal{R}|$ may both be much smaller than $C_\Pi \ln |\Pi|$, or even the improved preference distillation guarantee $C_0 \ln |\Pi|$ (24).

Astute readers may wonder: How can reverse KL avoid $C_\Pi$ (or $C_0$) while preference distillation does not, even though they both leverage the reward model? The reason is that the *policy learning*

---

[4]Experimental details: we set both the vocabulary and context sizes to 10 and instantiate $\pi^*, \pi_0$ as random categorical distributions. For each method and $n$, we report the result from the best $\beta$ from a wide range.

Table 2: **Results on TL;DR dataset with Pythia 1.4B and 2.8B**. Win-rate is evaluated by GPT-4 and reward model (RM) score evaluated by the trained reward model. For PyThia-1.4B, we report a single run due to the high computational cost of training both SFT and reward models from scratch, whereas for PyThia-2.8B we leverage publicly available models and report mean/std over three random seeds.

| Model size | Algorithm | Win-rate (%) ($\uparrow$) | RM score ($\uparrow$) | $\mathrm{KL}(\pi\|\pi_0)(\downarrow)$ |
|---|---|---|---|---|
| 1.4B | DPO | 45.0 | 1.03 | 33.46 |
| | PMLE | **46.0** | **1.12** | **19.60** |
| | REBEL | 59.5 | 2.60 | **31.67** |
| | Preference distillation | **62.5** | **2.61** | 33.31 |
| | RLHF | 60.0 | **2.74** | 24.41 |
| | Reverse KL | **61.5** | 2.73 | **23.91** |
| 2.8B | DPO | $47.3_{\pm 1.01}$ | $\mathbf{2.66}_{\pm 0.03}$ | $64.35_{\pm 0.68}$ |
| | PMLE | $\mathbf{49.1}_{\pm 1.43}$ | $2.38_{\pm 0.04}$ | $\mathbf{31.42}_{\pm 0.79}$ |
| | REBEL | $71.0_{\pm 1.26}$ | $2.86_{\pm 0.02}$ | $\mathbf{25.67}_{\pm 0.61}$ |
| | Preference distillation | $\mathbf{73.8}_{\pm 1.47}$ | $\mathbf{2.91}_{\pm 0.03}$ | $28.38_{\pm 0.82}$ |
| | RLHF | $72.0_{\pm 0.67}$ | $2.95_{\pm 0.05}$ | $\mathbf{24.94}_{\pm 1.12}$ |
| | Reverse KL | $\mathbf{73.0}_{\pm 0.93}$ | $\mathbf{3.03}_{\pm 0.04}$ | $25.91_{\pm 0.94}$ |

*step* of preference distillation still relies on response pairs $(a^+, a^-)$ sampled from $\mu$ (or $\pi_0$), unlike reverse KL which uses responses for the lightweight *reward modeling step* only. Moreover, bounding forward KL instead of reverse KL (even though the objective is derived from minimizing reverse KL) allows us to avoid comparing the coverage of $\hat{\pi}$ against $\mu$, which is not guaranteed by Definition 3. Nevertheless, the forward and reverse KL error may still be compared (however suffering a constant exponential in $R$), as we show in Proposition 15 in the appendix.

An alternative method is to directly optimize forward KL: $\arg\min_{\pi \in \Pi} \sum_{(x,\cdot,\cdot) \sim D_n} \mathrm{KL}(\widetilde{\pi}(x)\|\pi(x))$. However, this is computationally intractable since it requires evaluating normalization constants or sampling from unnormalized distributions. We defer detailed discussion to Appendix B.

## 6 EXPERIMENTS

In this section, we present empirical results showing that alignment via distribution learning yields strong performance in practice. Specifically, we validate our proposed methods by comparing them against their well-established baselines – DPO, RLHF, and REBEL – on a range of language tasks.

### 6.1 TL;DR SUMMARIZATION

We focus on the TL;DR summarization task (Stiennon et al., 2020), largely adhering to the training procedure detailed by Gao et al. (2024); Song et al. (2024).

**Setup.** We use Pythia-1.4B and Pythia-2.8B (Biderman et al., 2023). The details of SFT policy and reward model are provided in Appendix. We train DPO and PMLE on preference datasets annotated labels, whereas the online RL approaches use the dataset containing only human reference responses. All models are initialized with the SFT model prior to alignment, and RLHF and reverse KL minimization are optimized using PPO.

**Evaluation.** For each algorithm, we measure both the reward score assigned by our learned reward model and the KL divergence from the reference model, $\mathrm{KL}(\pi\|\pi_0)$. To evaluate the quality of model-generated responses, we use GPT-4 to compare them against human reference responses, calculating the win-rate. This win-rate is computed over 600 randomly selected samples, which corresponds to roughly 10% of the test set. We describe additional details and experiments in Appendix D.

**Results.** For both Pythia-1.4B and Pythia-2.8B in Table 2, our distribution-learning objectives – PMLE, preference distillation, and reverse KL – consistently outperform their respective baselines

Table 3: **Alignment Tax**. Performance comparison across academic benchmarks.

| Model | MMLU (5-shot) | GSM8K (5-shot) | ARC (25-shot) | WINOG (5-shot) | TRUTH (0-shot) | HELLA (10-shot) | Avg. | RM Score |
|---|---|---|---|---|---|---|---|---|
| LLaMA-3-8B-Instruct | 65.8 | 75.3 | 62.0 | 75.5 | 51.7 | 78.7 | 68.1 | - |
| REBEL-LLaMA-3 | 65.6 | 76.5 | 61.9 | 75.6 | 51.4 | 78.6 | 68.2 | 2610 |
| Distill-LLaMA-3 | 65.7 | 76.5 | 62.2 | 75.3 | 51.5 | 78.7 | **68.3** | **2697** |

in terms of win-rate, which serves as the most direct measure of LM quality. Additionally, since PMLE implements a KL regularizer with online data, it achieves much lower KL term compared to DPO, which solely relies on the offline dataset; this finding aligns with the results reported by Song et al. (2024). As for RLHF and REBEL, both methods use the same KL penalty for each experiment, naturally leading to similar $\mathrm{KL}(\pi\|\pi_0)$ values. Overall, our experiments demonstrate that the algorithms derived from our assumption (2) can match or exceed baselines on practical tasks.

## 6.2 GENERAL CHAT

Prior studies (Noukhovitch et al., 2023; Lin et al., 2024) suggest that aligning LLM with RLHF can incur an *alignment tax*, where models forget some of their pretrained capabilities, leading to performance degradation on standard benchmarks. We hypothesize that our framework mitigates this issue more effectively than conventional reward maximization. To this end, we examine the performance of our method across multiple language tasks when trained on a more general chat dataset while maintaining the downstream task performance. In particular, we compare our best-performing approach from the previous section, preference distillation, against its counterpart algorithm, REBEL.

**Setup.** We train LLaMA-3-8B-Instruct (Grattafiori et al., 2024) as our base model on the Ultra-FeedBack dataset (Cui et al., 2023), using Eurus-RM-7B (Yuan et al., 2024) as the reward model. We provide further experimental details such as hyperparameter settings in Appendix D.

**Evaluation.** Building on earlier work, we measure the alignment tax, i.e., the extent of performance deterioration using the Open LLM leaderboard (Beeching et al., 2023) as metrics, a widely adopted criteria for LLM evaluation. In particular, we focus on MMLU (Hendrycks et al., 2021), GSM8K (Cobbe et al., 2021), ARC challenge (Clark et al., 2018), Winogrande (Sakaguchi et al., 2021), TruthfulQA (Lin et al., 2022), and HellaSwag (Zellers et al., 2019) as done in Gao et al. (2024); Chen et al. (2025); Xie et al. (2025a).

**Results.** Table 3 presents the results on academic benchmarks. Preference distillation exhibits a similar alignment tax compared to REBEL while achieving higher reward scores (last column). This show that preference distillation can generate more preferred responses with similar alignment tax. This finding lends further support to our preference assumption (3) in conjunction with our discussion indicating that a learned RM score can serve as an indirect proxy for the underlying preference distribution (Section 4). We also include evaluations on MT-Bench/AlpacaEval in Appendix D.

## 7 CONCLUSION

In this paper, we have studied the significance of making clear assumptions about the target model $\pi^*$ and its relationship to observed preferences, a perspective we found underexplored in existing literature. By formulating alignment as distribution learning based on our explicit modeling assumption (3), we naturally derived three novel alignment methods: PMLE, preference distillation, and reverse KL. We have shown that these approaches correct and generalize existing methods in a principled manner, and demonstrated strong convergence guarantees and empirical performance.

Our work opens several important directions for future research. First, it would be interesting to compare mode-seeking and mode-covering objectives in terms of response quality across various domains. Second, our framework can be extended to incorporate alternative divergence metrics or other preference modeling assumptions. Finally, the constants in our upper bounds could be improved; in particular, the exponential dependence on $R$ might be removed by considering the KL regularizer and assuming $\pi_0$ is sufficiently close to $\pi^*$.

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

# Appendix

## Table of Contents

## A  RELATED WORK

**Preference optimization with RL.**    A widely adopted paradigm in preference optimization is Reinforcement Learning from Human Feedback (RLHF). In this framework, one first trains a reward model–effectively serving as a classifier–on a preference dataset collected from human annotators, and subsequently leverages the learned reward model to run RL algorithms such as PPO (Christiano et al., 2017; Ziegler et al., 2019). RLHF and its variants have been instrumental in training prominent LLMs such as ChatGPT (OpenAI, 2022), and have achieved remarkable success across diverse applications such as text summarization, question answering, instruction following, and text-to-image generation (Stiennon et al., 2020; Nakano et al., 2022; Ouyang et al., 2022; Lee et al., 2023; Liang et al., 2024). We point the interested reader to Kaufmann et al. (2024) for a recent dedicated survey on RLHF.

**Without RL and without a reward model.**    Direct Preference Optimization (DPO) dispenses with an explicit reward model by treating the log-ratio of each preference pair as a training signal and directly training the policy with a single contrastive cross-entropy loss (Rafailov et al., 2023). Such an RL-free objective was shown to match PPO-based RLHF without requiring a reward model, value network, or on-policy sampling, and has led to variants such as distilled DPO (Tunstall et al., 2024), Cal-DPO (Xiao et al., 2024), diffusion DPO (Wallace et al., 2023), ΨPO (Azar et al., 2024), SLiC/SLiC-HF (Zhao et al., 2023b;a), GPO (Tang et al., 2024), $\chi$PO (Huang et al., 2025), R-DPO (Park et al., 2024), ODPO (Amini et al., 2024), SimPO (Meng et al., 2024), RRHF (Yuan et al., 2023), KTO (Ethayarajh et al., 2024), ORPO (Hong et al., 2024), and many more.

At the same time, such direct optimization from preference labels has been noted to underperform along some dimensions compared to conventional RLHF. One challenge stems from relying exclusively on an offline dataset, which can induce out-of-distribution responses. This is likely due to insufficient on-policy interaction during training (Song et al., 2024). Some hybrid approaches have been proposed to overcome this issue: iterative DPO performs iterative training with labeled online preferences (Liu et al., 2024), HyPO combines offline data for preference optimization and online data for KL regularization (Song et al., 2024), and online DPO utilizes fast and slow chasing to simulate competition (Qi et al., 2024).

**Without RL but with a reward model.**    Another prominent method of preference optimization is reward distillation. This line of work aims to distill information on a reward model's preferences directly into the policy. As discussed in Section 4, the REBEL objective (Gao et al., 2024) regresses

the log-ratio of the likelihoods of two responses on the reward difference using a simple squared-loss objective, which is repeated with batches of on-policy responses. Reward distillation from Fisch et al. (2025) can be seen as a simplified version of REBEL where we only use the responses from the preference dataset. DRDO learns a reward model and policy in one pass by jointly matching oracle rewards while also learning human preferences (Nath et al., 2025). Finally, Zhang et al. (2025) develops an LLM distillation pipeline to distill both data and rewards.

**Theoretical analyses of preference optimization.** Zhan et al. (2024) studies offline preference-based RL with an MLE-based reward model similar to ours, but only obtains guarantees in terms of maximizing the policy value. Xie et al. (2025a) proposes an exploratory version of DPO which is shown to achieves $\widetilde{O}(\sqrt{T})$ regret with a favorable coverage parameter. Zhang et al. (2024) proposes an online direct alignment algorithm which also attains $\widetilde{O}(\sqrt{T})$ regret. Xiong et al. (2024) derives regret bounds for online and offline versions of RLHF under a linearly parametrized reward model; see also Foster et al. (2025) for a theoretical analysis of RL with linear-softmax policies. Cen et al. (2025) introduces VPO, a value-regularized DPO-type objective for both online and offline RLHF, and also prove regret bounds under linear rewards. The $\chi$PO algorithm is shown to attain optimal sample complexity, also in terms of regret, under a weaker single-policy concentrability (Huang et al., 2025).

The work of Agarwal et al. (2025) is most relevant to our paper, especially PMLE (Section 3): they develop a theoretical analysis of offline RLHF variants that minimize DPO-type objectives, and show a forward KL bound w.r.t. an optimal policy $\pi^*$. However, this formulation is not due to a distribution learning viewpoint but merely a byproduct of their strong realizability assumption (Assumption 3.2). Moreover, their upper bound has a square-root dependence on the excess risk $\varepsilon = L(\pi) - L(\pi^*)$, which when applied to our framework yields a statistical rate of $1/\sqrt{n}$. In contrast, we obtain an improved rate of $1/n$ with a more careful analysis in Appendix C.

## B ADDITIONAL REMARKS ON REVERSE KL

**Prior smoothing.** A key distinction from the standard RLHF objective lies in how our formulation balances reward maximization with *prior smoothing*. For an explicit comparison, we illustrate the effect of the additional entropy term for a toy alignment problem. Consider learning a $K$-categorical distribution on the simplex $\Delta_K = \{\mathbf{p} \in \mathbb{R}^d_{\geq 0} : \sum_{k=1}^K p_k = 1\}$, which can be viewed as a contextless language model with response length one and a vocabulary size of $K$. Suppose we are given a fixed vector $\mathbf{p}_0 \in \Delta_K$ as the reference model and a learned reward function $\hat{\mathbf{r}} = (r_1, \cdots, r_K)$. In the standard RLHF approach (1) with temperature $\beta + \gamma$, the optimal policy is given for all $k \in [K]$ by $\hat{p}_k^{\mathsf{RLHF}} \propto p_{0,k} \exp(\frac{r_k}{\beta+\gamma})$. In contrast, our reverse KL objective (16) can be rearranged as

$$\mathcal{L}_{\mathsf{RKL},\beta}(\mathbf{p}) = -\mathbf{p} \cdot \hat{\mathbf{r}} - \gamma H(\mathbf{p}) + \beta \, \mathrm{KL}(\mathbf{p}||\mathbf{p}_0)$$
$$= -\mathbf{p} \cdot \hat{\mathbf{r}} + (\beta + \gamma) \, \mathrm{KL}(\mathbf{p}||\mathbf{p}_0^\alpha) + \mathrm{const}.$$

where $\alpha := \frac{\beta}{\beta+\gamma}$, resulting in the policy $\hat{p}_k \propto p_{0,k}^\alpha \exp(\frac{r_k}{\beta+\gamma})$. The additional exponent $\alpha \in (0,1)$ acts to smooth the prior from $\mathbf{p}_0$ to $\mathbf{p}_0^\alpha$, allocating relatively more mass to actions with low initial probability. This boosts exploration especially for actions which were unlikely under the base policy, so that the estimated policy $\hat{\mathbf{p}}$ would not be too close to a degenerate distribution even if $\mathbf{p}_0$ is.

**Intractability of forward KL.** An alternative is to directly optimize the forward KL:

$$\arg \min_{\pi \in \Pi} \sum_{(x,\cdot,\cdot) \sim D_n} \mathrm{KL}(\widetilde{\pi}(x)\|\pi(x)).$$

Here, we are not using $(a^+, a^-)$, so the dependence on $\mu$ disappears and we will not pay for $C_\Pi$, similarly to Theorem 7. However, how do we compute the forward KL? Direct computation is untenable due to the sheer size of $\mathcal{A}$ in language models. Instead, one may attempt to sample from $\widetilde{\pi}(x)$ and perform stochastic optimization; however, such a sampling is not feasible because we only have access to the unnormalized version $\exp(\gamma^{-1}\hat{R}(x,\cdot))$. Another attempt is to use the fact that

$$\mathrm{KL}(\widetilde{\pi}(x)\|\pi(x)) = \mathbb{E}_{a \sim \pi(x)} \left[ \frac{\widetilde{\pi}(a \mid x)}{\pi(a \mid x)} \ln \frac{\widetilde{\pi}(a \mid x)}{\pi(a \mid x)} \right].$$

While we do not have to sample from $\widetilde{\pi}(x)$, we now have to evaluate the value of $\widetilde{\pi}(a \mid x)$, which, again, is intractable.

## C  THEORETICAL GUARANTEES

### C.1  AUXILIARY LEMMAS

We require the following basic results.

**Lemma 8.** *For all $a, b \in \mathbb{R}$ it holds that $|\sigma(a) - \sigma(b)| \geq \frac{1}{2}e^{-(|a| \vee |b|)}|a - b|$.*

*Proof.* Recall that $\sigma(z) = 1/(1 + \exp(-z))$ is the logistic sigmoid. $\sigma'$ is symmetric, so that

$$\sigma'(z) = \sigma'(|z|) = \frac{1}{1 + e^{|z|}}\frac{1}{1 + e^{-|z|}} \geq \frac{1}{1 + e^{|z|}} \geq \frac{1}{2}e^{-|z|}.$$

It suffices to assume $b > a$ due to symmetry. Then,

$$\sigma(b) - \sigma(a) = \int_a^b \sigma'(z)\,\mathrm{d}z \geq \int_a^b \frac{1}{2}e^{-|z|}\,\mathrm{d}z \geq \frac{1}{2}e^{-(|a| \vee |b|)}(b - a)$$

as desired. $\qquad\square$

The next two lemmas will allow us to convert between the expectation of the log ratio (i.e., KL divergence) and squared log ratio. Let us define the auxiliary function

$$\psi(z) := \frac{z - 1 - \ln z}{(\ln z)^2}.$$

**Lemma 9.** *For $r_{\max} > 1$, it holds for all $r \in (0, r_{\max}]$ that*

$$r - 1 - \ln r \leq (\frac{1}{2} \vee \psi(r_{\max}))(\ln r)^2 \leq \frac{r_{\max}}{(\ln r_{\max})^2}(\ln r)^2.$$

*Proof.* Define the auxiliary function

$$f(r) := \frac{1}{2}(\ln r)^2 - (r - 1 - \ln r).$$

For $r \in (0, 1)$, it holds that $f(1) = 0$ and $f'(r) = \frac{\ln r - r + 1}{r} < 0$. Thus, $f(r) > 0$ which implies

$$r - 1 - \ln r \leq \frac{1}{2}(\ln r)^2.$$

For $r \in [1, r_{\max}]$, it is easily checked that $\psi$ is nondecreasing on $(1, \infty)$ and thus

$$\frac{r - 1 - \ln r}{(\ln r)^2} = \psi(r) \leq \psi(r_{\max}) \leq \frac{r_{\max}}{(\ln r_{\max})^2},$$

as was to be shown. $\qquad\square$

**Lemma 10.** *For $r_{\min} > 0$, it holds for all $r \in [r_{\min}, \infty)$ that*

$$r - 1 - \ln r \geq \frac{1}{e(\ln r_{\min}^{-1} \vee 1)}(\ln r)^2.$$

*Proof.* The function $\psi$ defined in Lemma 9 extends to a nondecreasing continuous function on $(0, \infty)$ by setting $\psi(1) := \frac{1}{2}$. When $r \geq e^{-1}$, it follows that $\psi(r) \geq \psi(e^{-1}) = e^{-1}$.

When $r_{\min} \leq r < e^{-1}$, we use the fact that $\ln r \leq \frac{1}{1 - e^{-1}}(r - 1)$ to bound

$$\psi(r) \geq \frac{(1 - e^{-1})\ln r - \ln r}{(\ln r)^2} = \frac{1}{e \ln r^{-1}} \geq \frac{1}{e \ln r_{\min}^{-1}}.$$

$\qquad\square$

**Lemma 11** (Symmetrization inequality). *Let $D_n, \tilde{D}_n$ be two datasets of $n$ i.i.d. samples, $C(\pi, D_n)$ be any functional of a policy $\pi$ and dataset $D_n$, and $\hat{\pi} := \hat{\pi}(D_n)$ be any estimator computed from $D_n$. Then with probability $1 - \delta$, it holds that*

$$-\log \mathbb{E}_{\tilde{D}_n}[\exp(C(\hat{\pi}, \tilde{D}_n))] \leq -C(\hat{\pi}, D_n) + \ln(|\Pi|/\delta).$$

*Proof.* This is shown for example in the proof of Theorem 6 in Foster & Krishnamurthy (2021). $\square$

### C.2 PROOFS FOR SECTION 3

The following convergence bound for maximum likelihood estimators is mostly classical (Zhang, 2007; van de Geer, 2009); for completeness, we provide a brief proof following Theorem 6 of Foster & Krishnamurthy (2021).

**Proposition 12.** *Let $\hat{\pi} = \arg\min_{\pi \in \Pi} \mathcal{L}_{\mathsf{PMLE}}(\pi)$ with $\beta = 0$. Then, with probability at least $1 - \delta$,*

$$\mathbb{E}_{x \sim \mathcal{D}, a, b \sim \mu(x)} \left[ (\mathbb{P}_{\hat{\pi}}(a \succ b \mid x) - \mathbb{P}_*(a \succ b \mid x))^2 \right] \leq \frac{4 \ln(|\Pi|/\delta)}{n}.$$

*Proof.* Recall that each preference pair $(x, a^+, a^-)$ is collected by first sampling $a, b$ independently from $\mu(x)$ and setting $(a^+, a^-) = (a, b)$ with probability $\mathbb{P}_*(a \succ b \mid x)$. In other words, for the indicator $y = 1_{\{a^+ = a\}}$ such that $\mathbb{P}(y = 1) = \mathbb{P}_*(a \succ b \mid x)$, we can write

$$\mathcal{L}_{\mathsf{PMLE}}(\pi) = \frac{1}{n} \sum_{(x,a,b) \in D_n} -y \ln \mathbb{P}_\pi(a \succ b \mid x) - (1 - y) \ln \mathbb{P}_\pi(b \succ a \mid x),$$

where we have abused notation to write the sum over $(x, a, b)$ corresponding to each $(x, a^+, a^-)$ as a sum over $(x, a, b) \in D_n$. Define the quantity

$$C(\pi, D_n) = \frac{1}{2} \sum_{(x,a,b) \in D_n} y \ln \frac{\mathbb{P}_\pi(a \succ b \mid x)}{\mathbb{P}_*(a \succ b \mid x)} + (1 - y) \ln \frac{\mathbb{P}_\pi(b \succ a \mid x)}{\mathbb{P}_*(b \succ a \mid x)}$$

$$= \frac{n}{2}(\mathcal{L}_{\mathsf{PMLE}}(\pi^*) - \mathcal{L}_{\mathsf{PMLE}}(\pi))$$

and $\hat{\pi}$ as the minimizer of $\mathcal{L}_{\mathsf{PMLE}}(\pi)$ for $\pi \in \Pi$. It follows from Lemma 11 that

$$-\log \mathbb{E}_{\tilde{D}_n}[\exp(C(\hat{\pi}, \tilde{D}_n))] \leq -C(\hat{\pi}, D_n) + \ln(|\Pi|/\delta) \leq \ln(|\Pi|/\delta)$$

and

$$-\log \mathbb{E}_{\tilde{D}_n}[\exp(C(\hat{\pi}, \tilde{D}_n))]$$

$$= -n \log \mathbb{E}_{x \sim \mathcal{D}, a, b \sim \mu(x)} \mathbb{E}_{y|a,b,x} \left[ \left( \frac{\mathbb{P}_\pi(a \succ b \mid x)}{\mathbb{P}_*(a \succ b \mid x)} \right)^{y/2} \left( \frac{\mathbb{P}_\pi(b \succ a \mid x)}{\mathbb{P}_*(b \succ a \mid x)} \right)^{(1-y)/2} \right]$$

$$= -n \log \mathbb{E}_{x \sim \mathcal{D}, a, b \sim \mu(x)} \left[ \sqrt{\mathbb{P}_\pi(a \succ b \mid x) \mathbb{P}_*(a \succ b \mid x)} + \sqrt{\mathbb{P}_\pi(b \succ a \mid x) \mathbb{P}_*(b \succ a \mid x)} \right].$$

Writing $p_\pi = \mathbb{P}_\pi(a \succ b \mid x)$ and $p_* = \mathbb{P}_*(a \succ b \mid x)$ for simplicity, we further have

$$-\log \mathbb{E} \left[ \sqrt{p_\pi p_*} + \sqrt{(1 - p_\pi)(1 - p_*)} \right] \geq 1 - \mathbb{E} \left[ \sqrt{p_\pi p_*} + \sqrt{(1 - p_\pi)(1 - p_*)} \right]$$

$$= \mathbb{E} \left[ \frac{1}{2}(\sqrt{p_\pi} - \sqrt{p_*})^2 + \frac{1}{2}(\sqrt{1 - p_\pi} - \sqrt{1 - p_*})^2 \right]$$

$$= \mathbb{E} \left[ \frac{(p_\pi - p_*)^2}{2(\sqrt{p_\pi} + \sqrt{p_*})^2} + \frac{(p_\pi - p_*)^2}{2(\sqrt{1 - p_\pi} + \sqrt{1 - p_*})^2} \right]$$

$$\geq \frac{1}{4} \mathbb{E} \left[ (p_\pi - p_*)^2 \right],$$

which yields the desired bound. $\square$

*Proof of Theorem 4.* Our proof is partly inspired by Agarwal et al. (2025, proof of Theorem 3.6). The key difference is that their theorem relies on an assumption that the *population* loss of $\hat{\pi}$ is not too far away from that of $\pi^*$, which is rather strong. In contrast, our theorem provides an end-to-end guarantee. Furthermore, naively applying their theorem would result in an $1/\sqrt{n}$ rate rather than $1/n$. We obtain an improvement by applying Schulman's trick (Schulman, 2020) followed by Lemma 9. We elaborate more on this later in Remark 13.

Using Lemma 8 with the fact

$$\left|\gamma \ln \frac{\pi(a \mid x)}{\pi(b \mid x)}\right| = \left|\bar{R}(x, a) - \bar{R}(x, b)\right| \le 2\gamma R \,,$$

we can lower bound

$$\mathbb{E}_{x \sim \mathcal{D}, a, b \sim \mu(x)} \left[(\mathbb{P}_{\hat{\pi}}(a \succ b \mid x) - \mathbb{P}_*(a \succ b \mid x))^2\right]$$

$$\ge \frac{e^{-4\gamma R}}{4} \mathbb{E}_{x \sim \mathcal{D}, a, b \sim \mu(x)} \left[\left(\gamma \ln \frac{\hat{\pi}(a \mid x)}{\hat{\pi}(b \mid x)} - \gamma \ln \frac{\pi^*(a \mid x)}{\pi^*(b \mid x)}\right)^2\right]$$

$$= \frac{e^{-4\gamma R}}{4} \mathbb{E}_{x \sim \mathcal{D}, a, b \sim \mu(x)} \left[(\Delta \bar{R}_{\hat{\pi}}(x, a) - \Delta \bar{R}_{\hat{\pi}}(x, b))^2\right]$$

$$= \frac{e^{-4\gamma R}}{2} \mathbb{E}_{x \sim \mathcal{D}, a \sim \mu(x)} \left[\Delta \bar{R}_{\hat{\pi}}(x, a)^2\right]$$

where the last inequality is due to $\mathbb{E}[(X - Y)^2] = 2\mathbb{E}[(X - \mathbb{E}[X])^2]$ when $X$ and $Y$ are i.i.d. Thus, using Proposition 12, the difference in centered reward satisfies

$$\mathbb{E}_{x \sim \mathcal{D}, a \sim \mu(x)} \left[\Delta \bar{R}_{\hat{\pi}}(x, a)^2\right] \le 8e^{4\gamma R} \cdot \frac{\ln(|\Pi|/\delta)}{n}. \tag{18}$$

Define the normalizing factor

$$Z_\pi(x) := \sum_{a \in \mathcal{A}} \exp\left(\frac{1}{\gamma} \bar{R}_\pi(x, a)\right) = \exp\left(-\frac{1}{\gamma} \mathbb{E}_{a \sim \mu(x)}[R_\pi(x, a) \mid x]\right), \quad Z_* := Z_{\pi^*}$$

so that $\pi(a \mid x) = Z_\pi(x)^{-1} \exp(\gamma^{-1} \bar{R}_\pi(x, a))$. Due to Assumption 2, for all $\pi \in \Pi, x \in \mathcal{X}$ it holds that $|\mathcal{A}|e^{-R} \le Z_\pi(x) \le |\mathcal{A}|e^R$, so that

$$0 < \frac{\hat{\pi}(a \mid x)}{\pi^*(a \mid x)} = \frac{Z_*(x)}{Z_{\hat{\pi}}(x)} \exp\left(\frac{1}{\gamma} \Delta \bar{R}_{\hat{\pi}}(x, a)\right) \le e^{4R}. \tag{19}$$

Then, we bound the KL divergence between $\pi^*, \hat{\pi}$ using Schulman's trick (Schulman, 2020) followed by Lemma 9:

$$\mathbb{E}_{x \sim \mathcal{D}} \left[\mathrm{KL}(\pi^*(x) \| \hat{\pi}(x))\right] = \mathbb{E}_{x \sim \mathcal{D}, a \sim \pi^*(x)} \left[\ln \frac{\pi^*(a \mid x)}{\hat{\pi}(a \mid x)}\right]$$

$$= \mathbb{E}_{x \sim \mathcal{D}, a \sim \pi^*(x)} \left[\frac{\hat{\pi}(a \mid x)}{\pi^*(a \mid x)} - 1 - \ln \frac{\hat{\pi}(a \mid x)}{\pi^*(a \mid x)}\right]$$

$$\le \left(\frac{1}{2} \vee \psi(e^{4R})\right) \mathbb{E}_{x \sim \mathcal{D}, a \sim \pi^*(x)} \left[\left(\ln \frac{\hat{\pi}(a \mid x)}{\pi^*(a \mid x)}\right)^2\right]. \tag{20}$$

Extracting the normalization constants, we further have that

$$\mathbb{E}_{x \sim \mathcal{D}, a \sim \pi^*(x)} \left[\left(\ln \frac{\hat{\pi}(a \mid x)}{\pi^*(a \mid x)}\right)^2\right]$$

$$\le \mathbb{E}_{x \sim \mathcal{D}, a \sim \pi^*(x)} \left[2\left(\ln \frac{\hat{\pi}(a \mid x) Z_{\hat{\pi}}(x)}{\pi^*(a \mid x) Z_*(x)}\right)^2 + 2\left(\ln \frac{Z_*(x)}{Z_{\hat{\pi}}(x)}\right)^2\right]$$

$$= \frac{2}{\gamma^2} \mathbb{E}_{x \sim \mathcal{D}, a \sim \pi^*(x)} \left[\Delta \bar{R}_{\hat{\pi}}(x, a)^2\right] + 2\mathbb{E}_{x \sim \mathcal{D}} \left[\left(\ln \frac{Z_*(x)}{Z_{\hat{\pi}}(x)}\right)^2\right].$$

Using Definition 3 and (18), the first term is bounded as

$$\frac{2}{\gamma^2} \mathbb{E}_{x \sim \mathcal{D}, a \sim \pi^*(x)} \left[\Delta \bar{R}_{\hat{\pi}}(x, a)^2\right] \le \frac{2C_\Pi}{\gamma^2} \mathbb{E}_{x \sim \mathcal{D}, a \sim \mu(x)} \left[\Delta \bar{R}_{\hat{\pi}}(x, a)^2\right] \le \frac{16 C_\Pi e^{4\gamma R}}{\gamma^2} \cdot \frac{\ln(|\Pi|/\delta)}{n}.$$

For the second term, we first characterize an upper and lower bound on $\ln \frac{Z_{\hat{\pi}}(x)}{Z_*(x)}$. Using

$$1 = \mathbb{E}_{a \sim \pi^*(x)} \left[\frac{\hat{\pi}(a \mid x)}{\pi^*(a \mid x)}\right] = \frac{Z_*(x)}{Z_{\hat{\pi}}(x)} \mathbb{E}_{a \sim \pi^*(x)} \left[\exp\left(\frac{1}{\gamma} \Delta \bar{R}_{\hat{\pi}}(x, a)\right)\right],$$

we have

$$\ln \frac{Z_{\hat{\pi}}(x)}{Z_*(x)} = \ln \mathbb{E}_{a \sim \pi^*(x)} \left[ \exp\left( \frac{1}{\gamma} \Delta \bar{R}_{\hat{\pi}}(x, a) \right) \right] \geq \frac{1}{\gamma} \mathbb{E}_{a \sim \pi^*(x)}[\Delta \bar{R}_{\hat{\pi}}(x, a)]$$

where the last inequality is by Jensen's inequality. Moreover, using the inequality $e^x \leq 1 + x + \frac{e^A}{2}x^2$ valid for all $x \in (-\infty, A]$, we have

$$\ln \frac{Z_{\hat{\pi}}(x)}{Z_*(x)} = \ln \mathbb{E}_{a \sim \pi^*(x)} \left[ \exp\left( \frac{1}{\gamma} \Delta \bar{R}_{\hat{\pi}}(x, a) \right) \right]$$

$$\leq \mathbb{E}_{a \sim \pi^*(x)} \left[ \exp\left( \frac{1}{\gamma} \Delta \bar{R}_{\hat{\pi}}(x, a) \right) \right] - 1$$

$$\leq \frac{1}{\gamma} \mathbb{E}_{a \sim \pi^*(x)} \left[ \Delta \bar{R}_{\hat{\pi}}(x, a) \right] + \frac{e^{2R}}{2\gamma^2} \mathbb{E}_{a \sim \pi^*(x)} \left[ \Delta \bar{R}_{\hat{\pi}}(x, a)^2 \right].$$

Thus, we have

$$\left| \ln \frac{Z_{\hat{\pi}}(x)}{Z_*(x)} \right| \leq \left| \frac{1}{\gamma} \mathbb{E}_{a \sim \pi^*(x)}[\Delta \bar{R}_{\hat{\pi}}(x, a)] \right| + \frac{e^{2R}}{2\gamma^2} \mathbb{E}_{a \sim \pi^*(x)} \left[ \Delta \bar{R}_{\hat{\pi}}(x, a)^2 \right],$$

which implies, using $\forall x, y \in \mathbb{R}, (x + y)^2 \leq 2x^2 + 2y^2$,

$$\mathbb{E}_{x \sim \mathcal{D}} \left[ \left( \ln \frac{Z_*(x)}{Z_{\hat{\pi}}(x)} \right)^2 \right]$$

$$\leq \frac{2}{\gamma^2} \mathbb{E}_{x \sim \mathcal{D}} \left[ \left( \mathbb{E}_{a \sim \pi^*(x)} \left[ \Delta \bar{R}_{\hat{\pi}}(x, a) \right] \right)^2 \right] + \frac{e^{4R}}{2\gamma^4} \mathbb{E}_{x \sim \mathcal{D}} \left[ \left( \mathbb{E}_{a \sim \pi^*(x)} \left[ \Delta \bar{R}_{\hat{\pi}}(x, a)^2 \right] \right)^2 \right]$$

$$\leq \frac{2R^2 e^{4R} + 2}{\gamma^2} \mathbb{E}_{x \sim \mathcal{D}, a \sim \pi^*(x)} \left[ \Delta \bar{R}_{\hat{\pi}}(x, a)^2 \right] \qquad \text{(Jensen's inequality; Assumption 2)}$$

$$\leq \frac{16 C_\Pi (R^2 e^{4R} + 1)e^{4\gamma R}}{\gamma^2} \cdot \frac{\ln(|\Pi|/\delta)}{n} . \qquad \text{(by (18))}$$

Putting everything together, we conclude:

$$\mathbb{E}_{x \sim \mathcal{D}} \left[ \mathrm{KL}(\pi^*(x) \| \hat{\pi}(x)) \right]$$

$$\leq (\frac{1}{2} \vee \psi(e^{4R})) \left( \frac{16 C_\Pi e^{4\gamma R}}{\gamma^2} \cdot \frac{\ln(|\Pi|/\delta)}{n} + \frac{32 C_\Pi (R^2 e^{4R} + 1)e^{4\gamma R}}{\gamma^2} \cdot \frac{\ln(|\Pi|/\delta)}{n} \right)$$

$$= (\frac{1}{2} \vee \psi(e^{4R})) \frac{16(2R^2 e^{4R} + 3)C_\Pi e^{4\gamma R}}{\gamma^2} \cdot \frac{\ln(|\Pi|/\delta)}{n}.$$

We remark that by Lemma 9, the $\frac{1}{2} \vee \psi(e^{4R})$ term is further bounded above by $\frac{e^{4R}}{16R^2}$.

*Remark* 13. One of our key novelties is (20). In Agarwal et al. (2025), they use Cauchy-Schwarz to derive the bound

$$\mathbb{E}_{x \sim \mathcal{D}, a \sim \pi^*(x)} \left[ \ln \frac{\pi^*(a \mid x)}{\hat{\pi}(a \mid x)} \right] \leq \sqrt{\mathbb{E}_{x \sim \mathcal{D}, a \sim \pi^*(x)} \left[ \left( \ln \frac{\pi^*(a \mid x)}{\hat{\pi}(a \mid x)} \right)^2 \right]},$$

which introduces an extra square root compared to our derivation. Following their approach naively would lead to a $1/\sqrt{n}$ rate instead of $1/n$.

$\square$

## C.3 PROOFS FOR SECTION 4

*Proof of Theorem 6.* Up to constants, our distillation objective is equivalent to minimizing

$$\frac{1}{n} \sum_{(x, a^+, a^-) \in D_n} \mathrm{KL}\left( \mathrm{Bern}(\mathbb{P}_{\tilde{\pi}}(a^+ \succ a^- \mid x)) \| \mathrm{Bern}(\mathbb{P}_\pi(a^+ \succ a^- \mid x)) \right),$$

which can achieve zero loss since $\widetilde{\pi} \in \mathcal{P}_\gamma(\mathcal{R}) \subseteq \Pi$ is a valid solution. Thus, the solution $\hat{\pi}$ must satisfy

$$\mathbb{P}_{\widetilde{\pi}}(a \succ b \mid x) = \mathbb{P}_{\hat{\pi}}(a \succ b \mid x), \quad \forall (x, a, b) \in D_n$$

(recall that we use $(a, b)$ to denote the independent unlabeled responses). Defining the set

$$\mathcal{K} := \left\{ (\pi_1, \pi_2) \in \mathcal{P}_\gamma(\mathcal{R}) \times \Pi : \mathbb{E}_{x \sim \mathcal{D}, a, b \sim \mu(x)} \left[ | \mathbb{P}_{\pi_1}(a \succ b \mid x) - \mathbb{P}_{\pi_2}(a \succ b \mid x)| \right] > \varepsilon \right\},$$

it follows that

$$\mathbb{P}\left( (\widetilde{\pi}, \hat{\pi}) \in \mathcal{K} \right) = \sum_{(\pi_1, \pi_2) \in \mathcal{K}} \mathbb{P}(\widetilde{\pi} = \pi_1, \hat{\pi} = \pi_2)$$

$$\leq \sum_{(\pi_1, \pi_2) \in \mathcal{K}} \mathbb{P}\left( \mathbb{P}_{\pi_1}(a \succ b \mid x) = \mathbb{P}_{\pi_2}(a \succ b \mid x), \ \forall (x, a, b) \in D_n \right)$$

$$= \sum_{(\pi_1, \pi_2) \in \mathcal{K}} \mathbb{P}\left( \mathbb{P}_{\pi_1}(a \succ b \mid x) = \mathbb{P}_{\pi_2}(a \succ b \mid x) \right)^n$$

$$\leq \sum_{(\pi_1, \pi_2) \in \mathcal{K}} \left( 1 - \mathbb{E}\left[ | \mathbb{P}_{\pi_1}(a \succ b \mid x) - \mathbb{P}_{\pi_2}(a \succ b \mid x)| \right] \right)^n$$

$$\leq \sum_{(\pi_1, \pi_2) \in \mathcal{K}} (1 - \varepsilon)^n$$

$$\leq |\mathcal{K}|^2 \exp(-\varepsilon n).$$

Therefore $\mathbb{P}\left( (\widetilde{\pi}, \hat{\pi}) \in \mathcal{K} \right) \leq |\Pi|^2 \exp(-\varepsilon n)$, i.e.,

$$\mathbb{E}_{x \sim \mathcal{D}, a, b \sim \mu(x)} \left[ | \mathbb{P}_{\widetilde{\pi}}(a \succ b \mid x) - \mathbb{P}_{\hat{\pi}}(a \succ b \mid x)| \right] \leq \frac{2 \ln(|\Pi|/\delta)}{n}$$

with probability at least $1 - \delta$, and so

$$\mathbb{E}_{x \sim \mathcal{D}, a, b \sim \mu(x)} \left[ (\mathbb{P}_{\widetilde{\pi}}(a \succ b \mid x) - \mathbb{P}_{\hat{\pi}}(a \succ b \mid x))^2 \right] \leq \frac{2 \ln(|\Pi|/\delta)}{n} \tag{21}$$

as well. On the other hand, applying Proposition 12 to $\mathcal{P}_\gamma(\mathcal{R})$, we have

$$\mathbb{E}_{x \sim \mathcal{D}, a, b \sim \mu(x)} \left[ (\mathbb{P}_{\widetilde{\pi}}(a \succ b \mid x) - \mathbb{P}_*(a \succ b \mid x))^2 \right] \leq \frac{4 \ln(|\mathcal{R}|/\delta)}{n} \tag{22}$$

with probability at least $1 - \delta$. Hence by a union bound, it holds that, with probability at least $1 - \delta$,

$$\mathbb{E}_{x \sim \mathcal{D}, a, b \sim \mu(x)} \left[ (\mathbb{P}_{\hat{\pi}}(a \succ b \mid x) - \mathbb{P}_*(a \succ b \mid x))^2 \right] \leq \frac{4 \ln(2|\Pi|/\delta) + 8 \ln(2|\mathcal{R}|/\delta)}{n} .$$

Furthermore, by Lemma 8 it holds that

$$\left| \mathbb{P}_{\hat{\pi}}(a \succ b \mid x) - \mathbb{P}_*(a \succ b \mid x) \right| = \left| \sigma \left( \gamma \ln \frac{\hat{\pi}(a \mid x)}{\hat{\pi}(b \mid x)} \right) - \sigma \left( \gamma \ln \frac{\pi^*(a \mid x)}{\pi^*(b \mid x)} \right) \right|$$

$$\geq \frac{\gamma e^{-2\gamma R}}{2} \left| \ln \frac{\hat{\pi}(a \mid x)}{\hat{\pi}(b \mid x)} - \ln \frac{\pi^*(a \mid x)}{\pi^*(b \mid x)} \right|$$

$$= \frac{e^{-2\gamma R}}{2} \left| \Delta \bar{R}_{\hat{\pi}}(x, a) - \Delta \bar{R}_{\hat{\pi}}(x, b) \right|,$$

which implies that

$$\mathbb{E}_{x \sim \mathcal{D}, a \sim \mu(x)} \left[ (\Delta \bar{R}_{\hat{\pi}}(x, a))^2 \right] = \frac{1}{2} \mathbb{E}_{x \sim \mathcal{D}, a, b \sim \mu(x)} \left[ (\Delta \bar{R}_{\hat{\pi}}(x, a) - \Delta \bar{R}_{\hat{\pi}}(x, b))^2 \right]$$

$$\leq 2 e^{4\gamma R} \cdot \frac{4 \ln(2|\Pi|/\delta) + 8 \ln(2|\mathcal{R}|/\delta)}{n}. \tag{23}$$

Finally as in the proof of Theorem 4, we combine the bounds

$$\mathbb{E}_{x \sim \mathcal{D}, a \sim \pi^*(x)} \left[ \left( \ln \frac{\hat{\pi}(a \mid x)}{\pi^*(a \mid x)} \right)^2 \right]$$

$$\leq \frac{2}{\gamma^2} \mathbb{E}_{x\sim\mathcal{D},a\sim\pi^*(x)} \left[ \Delta\bar{R}_{\hat{\pi}}(x,a)^2 \right] + 2\,\mathbb{E}_{x\sim\mathcal{D}} \left[ \left( \ln \frac{Z_*(x)}{Z_{\hat{\pi}}(x)} \right)^2 \right]$$

and

$$\mathbb{E}_{x\sim\mathcal{D}} \left[ \left( \ln \frac{Z_*(x)}{Z_{\hat{\pi}}(x)} \right)^2 \right] \leq \frac{2R^2 e^{4R} + 2}{\gamma^2} \mathbb{E}_{x\sim\mathcal{D},a\sim\pi^*(x)} \left[ \Delta\bar{R}_{\hat{\pi}}(x,a)^2 \right]$$

along with (23) to conclude that

$$\mathbb{E}_{x\sim\mathcal{D}} \left[ \mathrm{KL}(\pi^*(x)\|\hat{\pi}(x)) \right] \leq (\frac{1}{2} \vee \psi(e^{4R})) \frac{16(2R^2 e^{4R} + 3)C_\Pi e^{4\gamma R}}{\gamma^2} \cdot \frac{\ln(2|\Pi|/\delta) + 2\ln(2|\mathcal{R}|/\delta)}{n}.$$

$\square$

**Theorem 14.** *Given a base policy $\pi_0$, denote by $C_0$ the smallest constant such that for every $\pi, \pi' \in \Pi$,[5]*

$$\mathbb{E}_{x\sim\mathcal{D},a\sim\pi^*(x)} \left[ (\Delta\bar{R}_\pi(x,a) - \Delta\bar{R}_{\pi'}(x,a))^2 \right] \leq C_0 \,\mathbb{E}_{x\sim\mathcal{D},a\sim\pi_0(x)} \left[ (\Delta\bar{R}_\pi(x,a) - \Delta\bar{R}_{\pi'}(x,a))^2 \right].$$

*The preference distillation estimate $\hat{\pi} = \arg\min_{\pi\in\Pi} \mathcal{L}_{\mathsf{Distill}}(\pi)$ with responses in $D_n$ generated from $\pi_0$ instead of $\mu$ satisfies*

$$\mathbb{E}_{x\sim\mathcal{D}} \left[ \mathrm{KL}(\pi^*(x)\|\hat{\pi}(x)) \right] \lesssim \frac{1}{\gamma^2} \cdot \frac{C_0 \ln(|\Pi|/\delta) + C_\mathcal{R} \ln(|\mathcal{R}|/\delta)}{n} \tag{24}$$

*with probability at least $1 - \delta$.*

*Proof.* In the proof of Theorem 6, (21) now holds with $\mu$ replaced by $\pi_0$, while (22) remains unchanged. It follows from Lemma 8 that

$$\frac{2\ln(|\Pi|/\delta)}{n}$$

$$\geq \mathbb{E}_{x\sim\mathcal{D},a,b\sim\pi_0(x)} \left[ (\mathbb{P}_{\widetilde{\pi}}(a \succ b \mid x) - \mathbb{P}_{\hat{\pi}}(a \succ b \mid x))^2 \right]$$

$$\geq \frac{\gamma^2 e^{-4\gamma R}}{4} \mathbb{E}_{x\sim\mathcal{D},a,b\sim\pi_0(x)} \left[ \left( \ln\frac{\hat{\pi}(a \mid x)}{\hat{\pi}(b \mid x)} - \ln\frac{\widetilde{\pi}(a \mid x)}{\widetilde{\pi}(b \mid x)} \right)^2 \right]$$

$$= \frac{e^{-4\gamma R}}{4} \mathbb{E}_{x\sim\mathcal{D},a,b\sim\pi_0(x)} \left[ (\Delta\bar{R}_{\hat{\pi}}(x,a) - \Delta\bar{R}_{\hat{\pi}}(x,b) - \Delta\bar{R}_{\widetilde{\pi}}(x,a) + \Delta\bar{R}_{\widetilde{\pi}}(x,b))^2 \right]$$

$$= \frac{e^{-4\gamma R}}{2} \mathbb{E}_{x\sim\mathcal{D},a\sim\pi_0(x)} \left[ (\Delta\bar{R}_{\hat{\pi}}(x,a) - \Delta\bar{R}_{\widetilde{\pi}}(x,a))^2 \right]$$

$$\geq \frac{e^{-4\gamma R}}{2C_0} \mathbb{E}_{x\sim\mathcal{D},a\sim\pi^*(x)} \left[ (\Delta\bar{R}_{\hat{\pi}}(x,a) - \Delta\bar{R}_{\widetilde{\pi}}(x,a))^2 \right],$$

where the last line uses the modified definition of the coverage coefficient $C_0$. Moreover, (22) implies

$$\mathbb{E}_{x\sim\mathcal{D},a\sim\pi^*(x)} \left[ (\Delta\bar{R}_{\widetilde{\pi}}(x,a))^2 \right] \leq C_\mathcal{R} \,\mathbb{E}_{x\sim\mathcal{D},a\sim\mu(x)} \left[ (\Delta\bar{R}_{\widetilde{\pi}}(x,a))^2 \right]$$

$$= \frac{1}{2} C_\mathcal{R} \,\mathbb{E}_{x\sim\mathcal{D},a,b\sim\mu(x)} \left[ (\Delta\bar{R}_{\widetilde{\pi}}(x,a) - \Delta\bar{R}_{\widetilde{\pi}}(x,b))^2 \right]$$

$$\leq 2C_\mathcal{R} e^{4\gamma R} \cdot \frac{4\ln(|\mathcal{R}|/\delta)}{n}.$$

By a union bound, it follows with probability at least $1 - \delta$ that

$$\mathbb{E}_{x\sim\mathcal{D},a\sim\pi^*(x)} \left[ (\Delta\bar{R}_{\hat{\pi}}(x,a))^2 \right]$$

$$\leq 2\,\mathbb{E}_{x\sim\mathcal{D},a\sim\pi^*(x)} \left[ (\Delta\bar{R}_{\hat{\pi}}(x,a) - \Delta\bar{R}_{\widetilde{\pi}}(x,a))^2 \right] + 2\,\mathbb{E}_{x\sim\mathcal{D},a\sim\pi^*(x)} \left[ (\Delta\bar{R}_{\widetilde{\pi}}(x,a))^2 \right]$$

$$\leq 8e^{4\gamma R} \cdot \frac{C_0 \ln(2|\Pi|/\delta) + 2C_\mathcal{R} \ln(2|\mathcal{R}|/\delta)}{n}.$$

The remainder of the proof proceeds similarly. $\square$

---

[5]This is slightly stronger than Definition 3, which can be retrieved by setting $\pi' = \pi^*$. We can weaken the definition to include only $\pi' \in \mathcal{R}$.

### C.4   Proofs for Section 5

*Proof of Theorem 7.* The first step of the argument is similar to the proof of Theorem 6, but with a stronger sup norm bound which is guaranteed due to realizability. Indeed, up to constants, the reverse KL objective is equivalent to

$$\hat{\pi} = \arg\min_{\pi \in \Pi} \frac{1}{n} \sum_{(x,\cdot,\cdot) \in D_n} \mathrm{KL}(\pi(x) \| \widetilde{\pi}(x)),$$

which achieves zero loss due to Assumption 5. Defining the set

$$K := \left\{ (\pi_1, \pi_2) \in \Pi \times \mathcal{P}_\gamma(\mathcal{R}) : \mathbb{E}_{x \sim \mathcal{D}} \left[ \sup_{a \in \mathcal{A}} \left( \ln \frac{\pi_1(a \mid x)}{\pi_2(a \mid x)} \right)^2 \right] > \varepsilon \right\},$$

it follows that

$$\mathbb{P} \left( \mathbb{E}_{x \sim \mathcal{D}} \left[ \sup_{a \in \mathcal{A}} \left( \ln \frac{\hat{\pi}(a \mid x)}{\widetilde{\pi}(a \mid x)} \right)^2 \right] > \varepsilon \right) = \sum_{(\pi_1, \pi_2) \in K} \mathbb{P}(\hat{\pi} = \pi_1, \widetilde{\pi} = \pi_2)$$

$$\leq \sum_{(\pi_1, \pi_2) \in K} \mathbb{P}(\pi_1(x) = \pi_2(x)), \ \forall x \in D_n)$$

$$= \sum_{(\pi_1, \pi_2) \in K} \mathbb{P}_{x \sim \mathcal{D}}(\pi_1(x) = \pi_2(x))^n.$$

Note that for any $(\pi_1, \pi_2) \in K$, it holds that $\pi_1(a \mid x)/\pi_2(a \mid x) \leq e^{4R}$ as in the proof of Theorem 4, so that

$$\sup_{a \in \mathcal{A}} \left( \ln \frac{\pi_1(a \mid x)}{\pi_2(a \mid x)} \right)^2 \leq 16R^2 \cdot 1_{\{\pi_1(x) \neq \pi_2(x)\}}, \quad \forall x \in \mathcal{X}.$$

This implies

$$\mathbb{P}_{x \sim \mathcal{D}}(\pi_1(x) = \pi_2(x)) = 1 - \mathbb{E}_{x \sim \mathcal{D}}[1_{\{\pi_1(x) \neq \pi_2(x)\}}] \leq 1 - \frac{\varepsilon}{16R^2},$$

and hence

$$\mathbb{P} \left( \mathbb{E}_{x \sim \mathcal{D}} \left[ \sup_{a \in \mathcal{A}} \left( \ln \frac{\hat{\pi}(a \mid x)}{\widetilde{\pi}(a \mid x)} \right)^2 \right] > \varepsilon \right) \leq |K| \left( 1 - \frac{\varepsilon}{16R^2} \right)^n \leq |\Pi|^2 \exp \left( -\frac{\varepsilon n}{16R^2} \right). \quad (25)$$

We now bound the forward KL divergence. Again applying Lemma 9,

$$\mathbb{E}_{x \sim \mathcal{D}} \left[ \mathrm{KL}(\pi^*(x) \| \hat{\pi}(x)) \right]$$

$$\leq (\frac{1}{2} \vee \psi(e^{4R})) \, \mathbb{E}_{x \sim \mathcal{D}, a \sim \pi^*(x)} \left[ \left( \ln \frac{\hat{\pi}(a \mid x)}{\pi^*(a \mid x)} \right)^2 \right]$$

$$\leq (1 \vee 2\psi(e^{4R})) \left( \mathbb{E}_{x \sim \mathcal{D}} \left[ \sup_{a \in \mathcal{A}} \left( \ln \frac{\hat{\pi}(a \mid x)}{\widetilde{\pi}(a \mid x)} \right)^2 \right] + \mathbb{E}_{x \sim \mathcal{D}, a \sim \pi^*(x)} \left[ \left( \ln \frac{\widetilde{\pi}(a \mid x)}{\pi^*(a \mid x)} \right)^2 \right] \right).$$

The first term can be bounded via (25). For the second term, repeating the derivation of Theorem 4 over the policy class $\mathcal{P}_\gamma(\mathcal{R})$ instead of $\Pi$, we obtain

$$\mathbb{E}_{x \sim \mathcal{D}, a \sim \pi^*(x)} \left[ \left( \ln \frac{\widetilde{\pi}(a \mid x)}{\pi^*(a \mid x)} \right)^2 \right] \leq \frac{16 C_\mathcal{R}(R^2 e^{4R} + 1) e^{4\gamma R}}{\gamma^2} \cdot \frac{\ln(|\mathcal{R}|/\delta)}{n}.$$

Putting everything together, we conclude:

$$\mathbb{E}_{x \sim \mathcal{D}} \left[ \mathrm{KL}(\pi^*(x) \| \hat{\pi}(x)) \right]$$

$$\leq (1 \vee 2\psi(e^{4R})) \left( 16R^2 \cdot \frac{\ln(2|\Pi|^2/\delta)}{n} + \frac{16 C_\mathcal{R}(R^2 e^{4R} + 1) e^{4\gamma R}}{\gamma^2} \cdot \frac{\ln(2|\mathcal{R}|/\delta)}{n} \right)$$

with probability at least $1 - \delta$. $\qquad\square$

**Proposition 15.** *It holds that*

$$\mathbb{E}_x[\mathrm{KL}(\hat{\pi}(x)\|\pi^*(x))] \le \frac{(4R \vee 1)e^{8R+1}}{R^2}\mathbb{E}_x[\mathrm{KL}(\pi^*(x)\|\hat{\pi}(x))].$$

*Proof of Proposition 15.* Denote the ratio $r = \frac{\pi^*(a|x)}{\hat{\pi}(a|x)}$ for brevity. Using the same argument as (19), we have $r \in [e^{-4R}, e^{4R}]$. Then, we have

$$r - 1 - \ln r \overset{(a)}{\le} \frac{e^{4R}}{R^2}(\ln r)^2 = \frac{e^{4R}}{R^2}\left(\ln\frac{1}{r}\right)^2 \overset{(b)}{\le} \frac{e^{4R}}{R^2}e(4R \vee 1)\left(\frac{1}{r} - 1 - \ln\frac{1}{r}\right),$$

where $(a)$ is by Lemma 9 and $(b)$ is by Lemma 10 with $r$ replaced by $1/r$. Thus,

$$\begin{aligned}
\mathrm{KL}(\hat{\pi}(x)\|\pi^*(x)) &= \mathbb{E}_{a\sim\hat{\pi}(x)}[r - 1 - \ln r] \\
&\le \frac{e^{4R}}{R^2}e(4R \vee 1)\mathbb{E}_{a\sim\hat{\pi}(x)}\left[\frac{1}{r} - 1 - \ln\frac{1}{r}\right] \\
&\le \frac{e^{4R}}{R^2}e(4R \vee 1)e^{4R}\mathbb{E}_{a\sim\pi^*(x)}\left[\frac{1}{r} - 1 - \ln\frac{1}{r}\right] \\
&= \frac{e^{4R}}{R^2}e(4R \vee 1)e^{4R}\mathrm{KL}(\pi^*(x)\|\hat{\pi}(x)).
\end{aligned}$$

$\square$

# D  EXPERIMENTAL DETAILS

In Section D.1, we describe the detailed settings for our toy experiments. In Section D.2, we provide implementation details on model card, hyperparameters, and compute resources on training and evaluating on the TL;DR dataset. In Section D.3, we provide details on our general chat experiments from Section 6 and also show additional results on MT-Bench and AlpacaEval 2.0.

## D.1  TOY EXPERIMENTS

We adopt a tabular setting with vocabulary and context sizes both equal to $10$. Under the preference model (2), we fix $\gamma = 0.5$. To keep the oracle policy $\pi^*$ and reference policy $\pi_0$ close, for each context $x \in \mathcal{X}$, we draw logits $\alpha^*, \alpha_0 \in \mathbb{R}^{10}$ with entries i.i.d. $\mathcal{N}(0, 0.1^2)$ and set $\pi^*(\cdot \mid x) = \mathrm{softmax}(\alpha^*)$ and $\pi_0(\cdot \mid x) = \mathrm{softmax}(\alpha^0)$. We then train polices by optimizing (1) and (16) and evaluate the forward KL $\mathbb{E}_x[\mathrm{KL}(\pi^*(\cdot \mid x)\|\pi(\cdot \mid x)]$. We vary the sample size $n \in \{2^5, \cdots, 2^{16}\}$, tuning the KL coefficient $\beta$ for both RLHF and RKL at each $n$, while keeping $\gamma = 0.5$ fixed for RKL.

## D.2  TL;DR SUMMARIZATION

**Dataset.**  We use the TL;DR dataset that is widely used in related literature (Gao et al., 2024; Song et al., 2024; Huang et al., 2024), publicly available[6]. We summarize the dataset statistics in Table 4. Note that DPO and PMLE are trained on the preference dataset which has preference labels, and other algorithms evaluate the policy based on human references since they utilize the online responses.

Table 4: TL;DR dataset statistics.

| Dataset | Train | Valid | Test |
|---|---|---|---|
| Human Reference | 117K | 64.5K | 6.55K |
| Preference | 92.9K | 83.8K | N/A |

---

[6]https://github.com/openai/summarize-from-feedback

**Models.** We use Pythia-1.4B[7] and Pythia-2.8B[8] (Biderman et al., 2023) as our pretrained models, using maximum context length $512$ and maximum generation length up to $53$ tokens. In order for training efficiency, we use LoRA (Low-Rank Adapter, Hu et al. (2022)) for alignment after full-parameter tuning the SFT model.

**Implementations.** We implement our three approaches (PMLE, reverse KL, preference distillation) on the top of a publicly available codebase[9]; preference distillation in particular is based on another publicly available code baseline[10]. For PMLE (Section 3), we implement the KL regularizer in (5) using the online responses described in Song et al. (2024). The DPO baseline takes about 3 hours and PMLE requires about 6 hours with 4 A100 40GB GPUs. Also, reverse KL and preference distillation, as well as their corresponding baselines RLHF and REBEL, takes about 2.5 days with 4 A100 40GB GPUs. Lastly, the win-rate is judged by GPT-4 using the `gpt-4` checkpoint (as of May 23rd, 2025).

---

**Algorithm 1** Preference Distillation (Sec. 4)

---

1: **Input:** (Learned) reward $\hat{R}$, policy class $\Pi$, preference generating distribution $\mu$, learning rate $\eta$, training dataset of prompts $\{x_i\}_{i=1}^n$.
2: **Initialize:**
3: **for** $t = 0, 1, \ldots, T - 1$ **do**
4:     Sample two responses from $a_1, a_2 \sim \mu(\cdot \mid x)$ for a given prompt $x \sim \mathcal{D}$ for all $x \in \{x_i\}_{i=1}^n$.
5:     Compute the probabilities with preference simulator by

$$\mathbb{P}_{\tilde{\pi}}(a_1 \succ a_2 \mid x) := \sigma\big(\hat{R}(x, a_1) - \hat{R}(x, a_2)\big)$$

$$\mathbb{P}_{\tilde{\pi}}(a_2 \succ a_1 \mid x) := \sigma\big(\hat{R}(x, a_2) - \hat{R}(x, a_1)\big)$$

6:     Compute the preference distillation loss $\mathcal{L}_{\mathsf{Distill},\beta}(\pi)$ using (11) and (12).
7:     $\pi_{t+1} \leftarrow \pi_t - \eta \nabla \mathcal{L}_{\mathsf{Distill},\beta}(\pi_t)$
8: **end for**

---

**Pseudocode.** Since the implementations of PMLE and reverse KL are straightforward from the corresponding DPO and RLHF baseline, we present the pseudocode for preference distillation for better understanding. As noted in Gao et al. (2024), the base distribution $\mu$ can also be $\pi_t$ in our pseudocode (Algorithm 1). Following the baseline code implementations of REBEL, we also sample online responses from the distribution $\pi_t$.

**Hyperparameters.** We adopt almost the same hyperparameters used in several studies (Huang et al., 2024; Gao et al., 2024; Song et al., 2024). For completeness, we summarize the hyperparameters used in our experiments in Table 5. Note that Gao et al. (2024) trains only a single epoch for RLHF and REBEL, but we cannot reproduce their results with just one epoch. Rather, following the implementation details (Huang et al., 2024), we consider the total episodes $10^6$ which corresponds to roughly about $8.5$ epochs. In this setting, we could reproduce the baseline results or obtain better results. Hence, reverse KL and preference distillation are also evaluated under this setup.

### D.3 GENERAL CHAT

**Dataset and Models.** In this experiment, we use the UltraFeedBack dataset (Cui et al., 2023), which is used in various baselines. We use LLaMA-3-8B-Instruct[11] as our base model and Eurus-RM-7B[12] as the reward model. One can use other public base models and reward models as well. Following Gao et al. (2024), we apply a length penalty $\Gamma$ (for responses exceeding maximum response length) with a KL regularizer to the reward function: $r(x, a) = \hat{R}(x, y) - \zeta\big(\ln \pi_{\theta_t}(a \mid x) - \ln \pi_0(a \mid x)\big)$.

---

[7]https://huggingface.co/EleutherAI/pythia-1.4b-deduped
[8]https://huggingface.co/EleutherAI/pythia-2.8b-deduped
[9]https://github.com/vwxyzjn/summarize_from_feedback_details
[10]https://github.com/ZhaolinGao/REBEL
[11]https://huggingface.co/meta-llama/Meta-Llama-3-8B-Instruct
[12]https://huggingface.co/openbmb/Eurus-RM-7b

Table 5: Hyperparameter configurations for TL;DR summarization tasks.

| Setting | Parameters | |
|---|---|---|
| SFT & RM | batch size: 64
learning rate: 3e-6 | schedule: cosine decay
train epochs: 1 |
| DPO | batch size: 64
learning rate: 3e-6
schedule: linear decay | train epochs: 1
$\beta$: 0.05 |
| PMLE | batch size: 512
learning rate: 1e-6
schedule: linear decay | train epochs: 1
$\beta$: 1e-5
$\gamma$: 1e-2 |
| REBEL | batch size: 512
learning rate: 3e-6
schedule: linear decay
total episodes: 1e6 | num epochs: 4
$\eta$: 1.0
kl coefficient: 0.05 |
| Preference Distillation | batch size: 512
learning rate: 3e-6
schedule: linear decay
total episodes: 1e6 | num epochs: 4
$\gamma$: 0.1
kl coefficient: 0.05 |
| RLHF (via PPO) | batch size: 512
learning rate: 3e-6
schedule: linear decay
total episodes: 1e6
num epochs: 4 | discount factor: 1
gae $\lambda$: 0.95
clip ratio: 0.2
value function coeff: 0.1
kl coefficient: 0.05 |
| Reverse KL (Sec. 5) | batch size: 512
learning rate: 3e-6
schedule: linear decay | total episodes: 1e6
kl coefficient: 0.05
entropy coefficient: 0.01 |
| LoRA Adapter
Config | r: 1024
$\alpha$: 2048 | dropout: 0.0
bias: False |
| Generation
Config | sampling: true
top k: 0.0
top p: 1.0 | min length: 53
max new tokens: 53
temperature: 0.1 (for DPO and PMLE) or 0.7 (others) |

**Implementation.** As in the TL;DR experiments, our implementation for preference distillation is based on Gao et al. (2024), which is publicly available. The total training time for LLaMA-3-8B-Instruct takes around 7 days on 4 A100 GPUs. In order to evaluate the model quality using MT-bench (Zheng et al., 2023) and AlpacaEval 2.0 (Dubois et al., 2024), we use the publicly available GitHub repositories.[13][14]

**Hyperparameters.** Similar to TL;DR experiments, we adopt the hyperparemeter configurations of Gao et al. (2024); for completeness, the full specification is presented in Table 6. Note that, due to the scale of experiments, we choose the best hyperparameter $\gamma$ used in TL;DR experiments. Consequently, the reported performance of preference distillation might be conservative, as more fine-grained hyperparameter tuning could yield further performance gains.

**Additional Results.** In addition to alignment tax in Section 6, we also include the general benchmark for evaluating LLMs: (i) MT-Bench (Zheng et al., 2023) and (ii) AlpacaEval 2.0 (Dubois et al., 2024) for quality analysis in Table 7. In Table 7, the MT-bench score of REBEL is slightly higher than that of preference distillation, but are very similar to each other. However, note that preference distillation is much better than REBEL baseline in terms of AlpacaEval 2.0 win-rate (including LC win-rate), which is known to have a higher Spearman correlation with Chatbot Arena (Dubois et al.,

---

[13]https://github.com/lm-sys/FastChat/tree/main/fastchat/llm_judge
[14]https://github.com/tatsu-lab/alpaca_eval

Table 6: Hyperparameter configurations for general chat experiments.

| Setting | Parameters | |
|---|---|---|
| REBEL | batch size: 32 | |
| | learning rate: 1e-7 | |
| | schedule: linear decay | |
| | train epochs: 1 | |
| | num epochs: 4 | |
| | $\eta$: 1.0 | |
| | $\zeta$: 0.5 | |
| | $\Gamma : -4$ | |
| Preference distillation | batch size: 32 | |
| | learning rate: 1e-7 | |
| | schedule: linear decay | |
| | train epochs: 1 | |
| | num epochs: 4 | |
| | $\beta$: 0.05 | |
| | $\gamma$: 0.1 | |
| | $\Gamma$: -4 | |
| Generation Config | sampling: true | min length: 1024 |
| | top k: 0.0 | max new tokens: 1024 |
| | top p: 1.0 | temperature: 0.5 |

Table 7: Quality analysis for general chat experiments.

| Models | MT-Bench Average | AlpacaEval 2.0 LC Win-rate | AlpacaEval 2.0 Win-rate |
|---|---|---|---|
| LLaMA-3-8B-Instruct | 8.10 | 30.50 | 30.50 |
| LLaMA-3-8B-REBEL | **7.89** | 31.25 | 31.68 |
| LLaMA-3-8B-Distill | 7.79 | **32.59** | **33.04** |

2024) than MT-bench. Taken together, these findings demonstrate the substantial promise of our preference distillation approach.

