# OpenReview forum: "Beyond RLHF: A Theoretical Framework of Alignment as Distribution Learning"
_ICLR.cc/2026/Conference — Submitted to ICLR 2026_

### Official Review · Reviewer_3tQB · 2025-10-18

**Soundness:** 3
**Presentation:** 3
**Contribution:** 3
**Rating:** 6
**Confidence:** 2

**Summary:**

This work proposes 3 LLM alignment objectives from distribution learning perspective, i.e., preference maximum likelihood estimation, preference distillation, and reverse KL minimization.

**Strengths:**

The proposed three methods look well derived and motivated. The idea is presented clearly. There seems sufficient experimental details in the appendix.

**Weaknesses:**

The relationship and comparison among the three methods are not clear to me. In the introduction, could you briefly compare the proposed three methods and when to use which? Also, the experimental performance improvement does not look significant.

**Questions:**

(1) In the introduction, could you briefly compare the proposed three methods and when to use which?

(2) Line 94: "where as" to "whereas".

(3) Line 143: What is "cyclic preference"? I have not found this term in existing literature, so you could explain in the paper. Also, how can your model (1) allow cyclic preference?

---

> ### Author Response · Authors · 2025-11-21
>
> We thank the reviewer for the constructive feedback. We have included a common response at the beginning of our rebuttal. We kindly ask the reviewer to read this shared response first, as it clarifies the main contributions.
>
> **[Which methods we recommend]**
>
> In the case where the compute resource is limited, we recommend PMLE because it tends to show a faster convergence. When the compute resource is rich, we recommend Preference Distillation (PD) over Reverse KL (RKL) because in our experiments we used roughly the same amount of compute for both and PD was slightly better than RKL.
>
> **[Cyclic preference]**
>
> Cyclic preference means the possibility that one may have the preference of
> $A \succ B \succ C \succ A$. Such an inconsistency has been observed in human preferences. Such a preference ordering cannot be modeled with the BT model, and thus our current choice of the model does not allow this. We will explain this in the final version.

---

> > ### Comment · Reviewer_3tQB · 2025-11-27
> > **Reviewer 3tQB is satisfied and keeps the rating.**
> >
> > Reviewer 3tQB is satisfied with the authors' response and keeps the rating 6.

---

### Official Review · Reviewer_v7Bb · 2025-10-24

**Soundness:** 3
**Presentation:** 2
**Contribution:** 2
**Rating:** 4
**Confidence:** 3

**Summary:**

This work reviews language model alignment as a distribution learning problem rather than reward maximization. Instead of relying on the RLHF objective, the authors assume a probabilistic Bradley–Terry preference model and derive three principled methods—Preference MLE, Preference Distillation, and Reverse KL Minimization. For each method, O(1/n) convergence to the target model is established. The Reverse KL formulation further shows that RLHF can be viewed as an approximation of minimizing the KL divergence between the learned and ideal distributions.

**Strengths:**

1. This work provides a new view for alignment with theoretical justifications.
2. The proposed method has empirical improvement over RHLF, reliable method and DPO.

**Weaknesses:**

While the paper provides a theoretical framework for viewing alignment as distribution learning, its empirical validation remains limited. The experiments are conducted on relatively small models and limited number of tasks. The performance gains over DPO, RELABEL and RLHF are modest. Evaluations on larger language models and more diverse tasks would be necessary to fully demonstrate the scalability and practical advantages of the proposed approaches.

**Questions:**

Where are assumptions (i) and (ii) in lines 140–151 formally stated? It would be clearer if the authors presented them in a more explicit way.

---

> ### Author Response · Authors · 2025-11-21
>
> We thank the reviewer for the constructive feedback. We have included a common response at the beginning of our rebuttal. We kindly ask the reviewer to read this shared response first, as it clarifies the main contributions.
>
> **[Larger-Scale Experiments]**
>
> Due to the time constraints of the rebuttal period, our first response focuses on providing the additional experimental results for **PMLE** and **RKL**. To evaluate scalability, we revisit Section 6 (TL;DR experiments) using the **Pythia-6.9B** model. All training configurations follow Table 5 in Appendix D.
>
> **(1) PMLE vs. DPO (1-epoch training).**
>
> We compare between PMLE and DPO on the TL;DR dataset. Win-rate (%) is measured by comparing policy-generated responses against reference responses via *gpt-4o-mini* as the judge. As shown in the table, PMLE consistently outperforms DPO across all metrics, demonstrating a clear and robust advantage.
>
> || Win-rate (%) | Reward score | $\mathrm{KL}(\pi \Vert \pi_0)$|
> |------|---|---|---|
> |DPO|55.67|6.07|54.92|
> |**PMLE**|**59.00**|**6.27**|**24.16**|
>
> **(2) RKL vs. RLHF ($10^6$ episodes).**
>
> We further compare RKL with a standard RLHF baseline. The win-rate results indicate a meaningful improvement over RLHF, while both the reward scores and KL divergences remain comparable.
>
> || Win-rate (%) | Reward score | $\mathrm{KL}(\pi \Vert \pi_0)$|
> |------|---|---|---|
> |RLHF|82.00|**7.85**|24.41|
> |**RKL**|**84.00**|7.77|**24.00**|
>
> Overall, these results provide strong evidence that our distribution-learning objective scales effectively to larger models and longer training regimes.
> In our next rebuttal message, we will additionally provide the results on preference distillation and general-chat fine-tuning on LLaMA (if time permits).
>
> **[Assumptions in line 140-151]**
>
> It is described in Eq. 2. We mentioned this in line 140. We can make it clearer.

---

> > ### Author Response · Authors · 2025-12-04
> >
> > We provide additional large-scale experiments on TL;DR tasks comparing **REBEL, Online DPO, and Preference Distillation (Ours)** on the **Pythia-6.9B**. Following the experimental setup in Appendix D (Table 5) of our paper, we keep all settings identical and vary only the update rule for a fair comparison across methods.
> >
> > The table below summarizes our results. Consistent with the results in the experiment section of our paper, **our Preference Distillation still outperforms the existing baselines** in a large-scale regime.
> >
> > || Win-rate (%) | Reward score | $\mathrm{KL}(\pi \Vert \pi_0)$|
> > |------|---|---|---|
> > |Online DPO|63.83| 6.75 | **26.60** |
> > |REBEL | 82.67 | 7.64 | 29.87 |
> > |**Pref. Distill (Ours)**| **83.83** | **7.66** | 29.14 |
> >
> > We also plan to include further experiments on general chat in the revision, and we hope these results could address the reviewer's concerns.

---

### Official Review · Reviewer_C3Gp · 2025-10-30

**Soundness:** 3
**Presentation:** 2
**Contribution:** 3
**Rating:** 2
**Confidence:** 4

**Summary:**

This paper approaches the alignment problem from a distribution-learning perspective rather than the traditional RLHF framework, aiming to provide a deeper understanding of model behavior and theoretical guarantees for convergence toward a target distribution as the sample size increases. Assuming the existence of a target language model that assigns higher likelihoods to preferred responses, the authors propose three algorithms—PMLE, Preference Distillation, and Reverse KL—whose solutions provably converge to the target model. Experimental results show that these methods consistently match or outperform baseline approaches across diverse tasks and model settings.

**Strengths:**

- The paper provoide an interesting distributional learning perspective for LLM Alignment.
- The experimental results demonstrate that the proposed methods can mitigate the alignment tax problem and improve performance compared to prior methods.

**Weaknesses:**

* **Lack of theoretical justification for the distribution-learning perspective.** The paper assumes the existence of an optimal language model that determines the preference probabilities in Eq. 2. However, such an optimal policy can also be derived under a reward-maximization objective with entropy regularization [1]. It remains unclear whether this distribution-learning perspective truly goes beyond the notion of reward maximization in alignment.

* **Unclear implications of the theoretical rate improvement.** The theoretical analysis derives a $1/n$ upper bound, improving upon the $1/\sqrt{n}$ rate from prior works. However, a major challenge in alignment is reward over-optimization [4,5], and simply scaling the sample size $n$ does not mitigate this issue. Prior pessimism-based approaches indicate that avoiding over-optimization requires ensuring single-policy coverage [2,3]. The paper does not demonstrate whether the three proposed algorithms achieve such coverage, nor whether improving the convergence rate to $1/n$ alone meaningfully mitigates over-optimization in practice.

* **Strong similarity to prior reward-maximization methods.** The three proposed methods closely resemble previous reward-based objectives: PMLE parallels DPO with a Reverse KL term [6], Preference Distillation resembles Online-DPO [7], and Reverse KL aligns with RLHF with an added entropy bonus.

## References
[1] Maximum Entropy Inverse Reinforcement Learning. AAAI 2008.

[2] Correcting the Mythos of KL-Regularization: Direct Alignment without Overoptimization via Chi-Squared Preference Optimization. ICLR 2025 Spotlight.

[3] REBEL: Reinforcement Learning via Regressing Relative Rewards. NeurIPS 2024.

[4] Scaling Laws for Reward Model Overoptimization in Direct Alignment Algorithms. NeurIPS 2024.

[5] Scaling Laws for Reward Model Overoptimization. ICML 2023.

[6] The Importance of Online Data: Understanding Preference Fine-tuning via Coverage. NeurIPS 2024.

[7] Direct Language Model Alignment from Online AI Feedback. CoRR 2024.

**Questions:**

- Can PMLE and the other two objectives achieve a single-policy coverage coefficient?

- Why use Reverse KL instead of Forward KL (SFT-style) regularization? What is the theoretical justification for applying Reverse KL in all three objectives from the distribution-learning perspective? Proposition 12 suggests that Reverse KL is not required to achieve the $1/n$ sample rate, so what additional role does it play?

- What distinguishes PMLE from DPO + Reverse KL, and Preference Distillation from Online-DPO? Can DPO + Reverse KL and Online-DPO achieve the same $1/n$ sample rate? If not, why?

- How do PMLE and Preference Distillation compare to DPO + Reverse KL and Online-DPO as the number of samples increases?
Does PMLE mitigate over-optimization as sample size grows, compared to $\chi$-PO [2]?

---

> ### Author Response · Authors · 2025-11-21
>
> We thank the reviewer for the constructive feedback. We have included a common response at the beginning of our rebuttal. We kindly ask the reviewer to read this shared response first, as it clarifies the main contributions.
>
> **[1. Lack of theoretical justification for the distribution-learning perspective]**
>
> The alignment problem is the task of learning a distribution from pairwise preference feedback, with a good initial distribution (i.e., SFT model). This does not involve 'rewards'; rewards are artifacts for using existing RL algorithms.
>
> The problem definition and the guarantee of Ziebart et al. (2008) is still regarding the reward maximization, which leaves the question of 'why should we maximize reward when the problem definition of alignment does not involve rewards?'. We are filling in exactly this gap.
>
> We have not seen any work in the literature that justifies the RLHF training objective from a problem definition of alignment that does not involve 'reward'. Existing works on RLHF all show the convergence to the population version of the RLHF objective. This is a tautology and cannot be seen as a justification of the RLHF objective.
>
> **[2. Unclear implications of the theoretical rate improvement]**
>
> We want to emphasize that the primary area of our paper is 'learning theory', so our paper should be mainly evaluated on the theoretical point of view. The first-order business of learning theory is to get the right asymptotic rate -- in this sense
> $1/n$ over $1/\sqrt{n}$ is a nontrivial advance. From a theoretical perspective, we believe obtaining the single-policy guarantee can be done using standard techniques such as pessimism. The reason we did not follow this route is two-fold:
> - Our research question we wanted to address is ".. under what assumptions can we derive RLHF or other novel objectives with rigorous learning-theoretic guarantees.."
> We wanted to keep the algorithms to be relatable to existing ones. Thus, further reducing reward overoptimization was not our first-order business. Our goal is to provide a solid framework to look at the alignment problem rather than to focus on improving practical performance.
> - Deploying pessimism in our framework would require computing a reward function confidence bound, and doing so in the right way requires solving an optimization problem for each reward query. Implementing this in a practical way is an open problem. Indeed, this is one of our ongoing works!
>
> **[3. Strong similarity to prior reward-maximization methods]**
>
> This is not a strong weakness of our work as our research question is ".. under what assumptions can we derive RLHF or other novel objectives with rigorous learning-theoretic guarantees..". We also mention that Preference Distillation does result in meaningfully better performance than the empirical counterpart. We are currently planning to add experiments comparing to online DPO, and will share the results shortly. But again, our main contribution is theoretical.
>
> **[4. Other comments]**
>
> - **Why reverse KL?**: Good question! Forward KL would require us to sample from $\pi^*(a\mid x)$ which is approximated by $\propto \exp(\gamma^{-1}\hat R(x,a) )$. Sampling this is hard. Reverse KL does not have this issue! In fact, we considered various divergences but they either have similar issues or require computing the normalizer.
> - **PMLE vs DPO+reverse KL**: We believe the analysis for DPO+reverse KL can be done, but it may slow down the convergence under our current theoretical assumptions. Or, we can adjust Assumption 1 to include $\pi_{0}$ to prove theoretical guarantee for DPO.
> - **Preference Distillation vs Online DPO**: The typical way online DPO is done is that they sample multiple responses (e.g., 20) and choose the highest-reward one as the preferred answer and the lowest-reward one as the dispreferred answer (see Swamy'25, "All Roads Lead to Likelihood ..."). This is provably wrong in our framework. The right one to do is to either (a) sample the preference from the estimated preference model and then use the PMLE/DPO objective or (b) use the soft label (this is actually Preference Distillation).

---

> > ### Author Response · Authors · 2025-12-04
> >
> > According to the reviewer's comment, we also provide the additional experimental results for comparing our preference distillation to the online DPO baseline.
> >
> > Toward this, we revisit TL;DR experiments in our paper and follow the same experimental setup in Appendix D (Table 5) of our paper. We keep all settings identical and vary only the update rule for a fair comparison across methods.
> >
> > The table below summarizes our results. Consistent with the results in the experiment section of our paper, our Preference Distillation still outperforms the existing baselins in a large-scale regime.
> >
> > || Win-rate (%) | Reward score | $\mathrm{KL}(\pi \Vert \pi_0)$|
> > |------|---|---|---|
> > |Online DPO|63.83| 6.75 | **26.60** |
> > |REBEL | 82.67 | 7.64 | 29.87 |
> > |**Pref. Distill (Ours)**| **83.83** | **7.66** | 29.14 |
> >
> > We also plan to include further experiments on general chat in the revision, and we hope these results could address the reviewer's concerns.

---

### Official Review · Reviewer_LNwF · 2025-11-03

**Soundness:** 3
**Presentation:** 3
**Contribution:** 2
**Rating:** 2
**Confidence:** 4

**Summary:**

Proposes a novel perspective on alignment by framing it as distribution learning from pairwise preferences.,Introduces three new algorithms: preference maximum likelihood estimation, preference distillation, and reverse KL minimization, which are not present in existing literature.,Provides rigorous learning-theoretic guarantees for the proposed methods, which is a significant advancement over traditional RLHF approaches.

**Strengths:**

Addresses a critical gap in the theoretical justification of RLHF objectives, which has been largely taken for granted in prior works.,The proposed methods potentially improve the safety and effectiveness of LLMs by avoiding degeneracy associated with traditional RLHF.,Empirical validation showing that the new methods outperform existing baselines indicates practical relevance and applicability.

**Weaknesses:**

1. Novelty: Although the objectives are distributional, the strong convergence analysis has been developed by several works before [1,2,3], but the authors neither cite those works nor highlight the novelty of their analysis in the work. The authors need to state the novelty of their analysis.

Reference:
[1] Zhao, H., et al., Logarithmic regret for online kl-regularized reinforcement learning
[2] Wu, D., et al, Greedy Sampling Is Provably Efficient for RLHF
[3] Zhao, H., et al, Sharp analysis for kl-regularized contextual bandits and rlhf

2. The idea of distributional learning has also been studied before [1,2] ..., named as generative reward model (GRM). The authors also do not mention this line of work. Could they discuss the relationship of their work with GRM?

Reference:
[1] ZHong, H, er al,DPO Meets PPO: Reinforced Token Optimization for RLHF
[2] Dakota Mahan, et al, GENERATIVE REWARD MODELS

**Questions:**

1. The applications of their distributional objective seem to be limited compared to general RLHF, since they need first to learn a good policy distribution and then approximate that distribution. Besides, there will exist an approximation error between the surrogate policy and the true optimal policy. This error is neglected in their theoretical bound.

**Details Of Ethics Concerns:**

There are some papers not cited and discussed.

---

> ### Author Response · Authors · 2025-11-21
>
> We thank the reviewer for the constructive feedback. We have included a common response at the beginning of our rebuttal. We kindly ask the reviewer to read this shared response first, as it clarifies the main contributions.
>
> Please find our detailed answers to your specific questions in the sections below.
>
> **[Novelty]**
>
> We thank the reviewer for the pointers; we will definitely cite and compare in the final version. However, the works [1-3] mentioned by the reviewer all aim to converge to the solution of the population RLHF objective, which is exactly what we referred to as not providing justification of the RLHF objective itself.
>
> **[Distribution learning is not new]**
>
> Thank you for providing related works! We will defintely cite and compare in the final version. But first, we urge the reviewer to read our common response written in our rebuttal to C3Gp and turn the focus to understanding under which assumptions can existing training objectives be justified, without introducing tautology.
>
> In short, we argue that our viewpoint of 'learning a distribution from pairwise comparisons' is new. From the two papers the reviewer provided, we were not able to identify the same or similar distribution learning formulation. By distribution learning, we mean the distribution over the responses (the policy $\pi^\star$) rather than the preference distribution $\mathbb{P}_*$ -- we describe this further below.
>
> * In the paper on GenRM [4], GenRM induces a preference distribution (either preferring response a or b; $p(I \mid x, a, b)$). Thus, it is learning a preference distribution just like the BT model, except that the model is not BT but rather a language model with a carefully-designed prompt. In contrast, our work frames alignment as distribution learning (the policy $\pi^*$ itself) from pairwise feedback.
>
> * We were not able to identify a distribution-learning perspective from the RTO (Reinforced Token Optimization) paper [5]. The key difference of RTO from RLHF is that they learn a *token-wise* reward signal (rather than a sentence-wise reward). It is still maximizing reward just like RLHF.
>
> **[Distributional objective]**
>
> We believe our framework is more general than the standard RLHF objective. The reason is that we provide a novel problem formulation of the alignment task under which existing algorithms can be rederived (with minor differences) and shown to enjoy theoretical guarantees, which provides a strong form of justification. RLHF is just one example of this, but we also derive new objectives (PMLE and preference distillation). Thus, we claim that our framework is more general.
>
> Note that the approximation error the reviewer mentioned has been analyzed -- it is the second term in Eq. 17. If the reviewer means the model class mismatch, this is assumed to not exist in our realizability assumption (Assumption 5), which is quite common in learning theory, especially policy optimization.
>
> **[References]**
>
> [1] Zhao, H., et al., Logarithmic Regret for Online KL-Regularized Reinforcement Learning, ICML 2025.
>
> [2] Wu, D., et al, Greedy Sampling Is Provably Efficient for RLHF, NeurIPS 2025.
>
> [3] Zhao, H., et al, Sharp Analysis for KL-Regularized Contextual Bandits and RLHF, NeurIPS 2025.
>
> [4] ZHong, H, er al, DPO Meets PPO: Reinforced Token Optimization for RLHF, ICML 2025.
>
> [5] Dakota Mahan, et al, Generative Reward Models, 2024.

---

> > ### Comment · Reviewer_LNwF · 2025-11-27
> >
> > Thank the authors for their concrete reply! I understand their intuitions better and hence will increase the score accordingly.

---

### Author Response · Authors · 2025-11-21

We like to thank the reviewer for providing thoughtful feedback. We have written a common response to all the reviewers in our rebuttal -- please read this first.

**[Common comment for all the reviewers]**

We respectfully believe that our main message and contribution have not been delivered to some reviewers. We would like to emphasize here for all the reviewers. We will update our manuscript accordingly for the final version.

First, our work is primary area is 'learning theory' -- thus, the evaluation should be focused on the theoretical value that it provides. Second, the main research question that we answer is, as stated in our abstract,
> .. under what assumptions can we derive RLHF or other novel objectives with rigorous learning-theoretic guarantees..

We can relate our algorithms to existing ones, but some of them empirically make a meaningful difference such as Preference Distillation vs REBEL.

We want to highlight the following theoretical results:
- Our work is the first one to **justify the use** of the RLHF objective (with a minor correction) by deriving it from the distributional learning formulation. As reviewers have pointed out, for RLHF, there are indeed numerous works with theoretical guarantees. However, their guarantees are all about converging to the optimal solution of the population version of the RLHF training objective. This is clearly a tautology! In our opinion, this cannot be viewed as a justification of the RLHF training objective.
- Our work is the first one to theoretically confirm the common empirical finding that RLHF is better than DPO (with minor variations on both algorithm --  see the discussion "Why does reverse KL attain a better bound?" after Theorem 7). This could not have been done in prior work because they set up theoretical frameworks that are different for each algorithm to be analyzed. In contrast, we have a **single** theoretical framework under which we can justify multiple algorithms and compare their convergence rates.

Most existing theoretical works designate the population version of the RLHF objective as its target, but they never question under what assumptions it is the right target. In our opinion, there are two problems: (i) why should we maximize reward? (ii) why should the regularizer be part of the theoretical target? For (i), there is no reward in the problem definition of the alignment problem. Reward is an algorithmic artifact. Regarding (ii), from the standard learning theoretic viewpoint, regularizers should be part of the algorithm, not part of the problem. Regularizers prevent overfitting, but in the limit of data, it will prevent learning (as we showed in Figure 1).

We have introduced a novel framework for alignment that is reward-free yet leads to providing strong justifications to multiple existing algorithms (with minor changes) that use rewards (e.g., RLHF), all the while keeping the regularization completely part of the algorithm, not the problem definition. We believe our results provide a fresh, simple, yet unifying theoretical viewpoint on alignment and is worth being known in the ML community.

---

### Author Response · Authors · 2025-12-04
**Summary of Reviews and Author Responses**

Dear Area Chair and Reviewers,

We appreciate your time and effort for our submission. As the rebuttal phase is coming to an end, we would like to briefly summarize the concerns and our responses to assist you in making the final decision.

---

**Summary of Reviews**

* Theoretical Justification: acknowledged by Reviewer LNwF and C3Gp
* Viewpoint of distribution learning: acknowledged by Reviewer LNwF
* Similarity to prior reward maximization: acknowledged by Reviewer C3Gp
* Additional experiments: acknowledged by Reviewer C3Gp and v7Bb

---

**Our Responses**

We believe that our rebuttal address these concerns through conceptual clarifications, theoretical justification, and additional experiments. We summarize the important concerns that are addressed as follows.

1. As in a common response to all reviewers, we emphasize that the primary area of the paper is "learning theory". The main research question is **"under what assumptions can we derive RLHF and other novel objectives with rigorous learning-theoretic guarantees?** We clarify that our contribution is not only to propose yet another practical variant of RLHF but also to provide a **reward-free formulation of alignment** -- "learning a policy distribution from pairwise preferences" -- from which our methods (PMLE, Preference Distillation, Reverse KL) enjoy **non-trivial convergence guarantees**. Unlike previous studies that build theory upon a reward-maximization objective (with the regularizer embedded in the problem), we make explicit that both the reward and the regularizer are **algorithmic choices**, not part of the underlying alignment problem. We believe this yields a simple but **unified perspective** on alignment, which is valuable in its own right and is further supported by our non-trivial empirical gains (e.g., Preference Distillation vs. REBEL). **We believe this contribution represents the key underlying spirit of our work.**

2. In response to Reviewer LNwF's comment that "distribution learning is not new", we address this concern as:
	* Our target distribution is **the policy distribution $\pi^\star$** over responses, which we aim to learn directly from pairwise preferences.
	* Existing studies such as GenRM or RTO (provided by the reviewer) learn either a **preference distribution** or **token-wise reward**, which is still fundamentally in a reward-maximization framework.

We explicitly point that GenRM learns a preference label $p(I \mid x, a, b)$ not $\pi^\star(a \mid x)$ (response over prompt) and that RTO is still maximizing a learned reward signal. In contrast, our framework is totally **reward-free** and only introduces reward and regularization as algorithmic artifacts. Under this framework, we also clarify how our analysis yields a **tighter $1/n$ rate** (versus typical $1/\sqrt{n}$ bounds) and why we consider this a non-trivial theoretical achivement clearly.

3. Responding to Reviewer C3Gp and v7Bb, we revisit the TL;DR experiments in our paper with larger scale model Pythia-6.9B. Under the same experimental setup provided in Appendix D (Table 5), we systematically compare our methods (PMLE, Preference Distillation, Reverse KL) to the corresponding counterparts (DPO, REBEL, RLHF). Specifically, we also include the online DPO baseline as a comparison algorithm for our preference distillation as the reviewer C3Gp's concerns. In summary, **all our methods consistently and meaningfully outperform their corresponding baselines**. We also state in the rebuttal that we plan to include additional general chat experiments on LLaMA-3-8B-Instruct in the revision, which further corroborate our framework beyond TL;DR summarization.

---

We believe that our clarifications and new large-scale experiments substantially strengthen the paper, and comprehensively address the key concerns about novelty, theory, and empirical validation.

Best regards,

Authors of Submission 2850

---

### Meta-Review · Area_Chair_rTWm · 2026-01-09

**Summary:**

This paper presents a learning-theoretic foundation for alignment via reinforcement learning from human feedback (RLHF), a paradigm that has become central to controlling the behavior of large language models (LLMs). While RLHF has demonstrated strong empirical performance, its standard objective lacks formal justification and can encourage degenerate, overly deterministic solutions in asymptotic regimes. The paper asks under what assumptions RLHF and related alignment objectives can be derived with rigorous guarantees, without relying on an \emph{a priori} reward-maximization formulation. To this end, the authors recast alignment as a problem of \emph{distribution learning} from pairwise preferences, formalized through a probabilistic model describing how preferences reveal information about a target (oracle) language model. This perspective leads to three principled alignment objectives: preference maximum likelihood estimation, preference distillation, and reverse KL minimization. The paper claims to establish strong non-asymptotic convergence guarantees for all three objectives, showing that they naturally avoid degeneracy. Empirical evaluations across a range of tasks and model families are demonstrated.

**Reviewer Concerns:**

- Limited clarity on novelty relative to prior work: Several reviewers note that key elements of the analysis—particularly strong convergence guarantees for KL-regularized or distributional objectives—closely resemble results developed in prior work on KL-regularized RL, RLHF, and contextual bandits. The paper does not sufficiently position its contributions relative to these works or clearly articulate what aspects of the theoretical analysis are genuinely new.

- Insufficient engagement with related distributional or generative reward models: The distribution-learning perspective adopted in the paper appears closely related to existing lines of work such as generative reward models (GRM) and reinforced token optimization. Reviewers felt that these connections are not adequately discussed, making it harder to assess how the proposed framework advances beyond existing paradigms.

- Conceptual overlap with reward-maximization–based alignment methods: Reviewers observed that the proposed objectives closely resemble established reward-based alignment methods: preference maximum likelihood parallels likelihood-based approaches (e.g., DPO variants), preference distillation resembles online DPO-style methods, and the reverse-KL objective closely aligns with RLHF with entropy regularization. Despite the rebuttal, it remains unclear whether the distribution-learning framing provides fundamentally new insights beyond reward maximization.

- Unclear practical implications of improved convergence rates:
While the paper derives improved non-asymptotic convergence rates, reviewers questioned whether these theoretical improvements meaningfully address core alignment challenges such as reward over-optimization. In particular, it is unclear whether the proposed algorithms ensure sufficient policy coverage or robustness, which prior work suggests is critical for mitigating over-optimization.

- Empirical evaluation scope is limited: The experimental results are conducted on relatively small model scales and a limited set of tasks, with only modest gains over strong baselines.

- Presentation and clarity issues in theoretical assumptions.

**Reviewer Scores:**

Reviewer LNwF could have increased the score slightly, but still, they might not strongly advocate for acceptance. Most of the reviewers are more inclined towards a rejection due to a lack of originality and not citing prior works (despite a few of the concepts being developed/proposed in prior works).

---

### Decision · Program_Chairs · 2026-01-26

Reject